# Live-imaging of endothelial Erk activity reveals dynamic and sequential signalling events during regenerative angiogenesis

Kazuhide S Okuda[1,2,3,4], Mikaela S Keyser[4], David B Gurevich[5,6], Caterina Sturtzel[7,8], Elizabeth A Mason[1,2,3], Scott Paterson[1,2,3,4], Huijun Chen[4], Mark Scott[4], Nicholas D Condon[4], Paul Martin[5], Martin Distel[7,8], Benjamin M Hogan[1,2,3,4]*

[1]Organogenesis and Cancer Program, Peter MacCallum Cancer Centre, Melbourne, Australia; [2]Sir Peter MacCallum Department of Oncology, University of Melbourne, Melbourne, Australia; [3]Department of Anatomy and Physiology, University of Melbourne, Melbourne, Australia; [4]Institute for Molecular Bioscience, The University of Queensland, St Lucia, St Lucia, Australia; [5]School of Biochemistry, Biomedical Sciences Building, University Walk, University of Bristol, Bristol, United Kingdom; [6]Department of Biology and Biochemistry, University of Bath, Claverton Down, Bath, United Kingdom; [7]Innovative Cancer Models, St Anna Kinderkrebsforschung, Children's Cancer Research Institute, Vienna, Austria; [8]Zebrafish Platform Austria for preclinical drug screening (ZANDR), Vienna, Austria

*For correspondence:
ben.hogan@petermac.org

Competing interests: The authors declare that no competing interests exist.

**Abstract** The formation of new blood vessel networks occurs via angiogenesis during development, tissue repair, and disease. Angiogenesis is regulated by intracellular endothelial signalling pathways, induced downstream of vascular endothelial growth factors (VEGFs) and their receptors (VEGFRs). A major challenge in understanding angiogenesis is interpreting how signalling events occur dynamically within endothelial cell populations during sprouting, proliferation, and migration. Extracellular signal-regulated kinase (Erk) is a central downstream effector of Vegf-signalling and reports the signalling that drives angiogenesis. We generated a vascular Erk biosensor transgenic line in zebrafish using a kinase translocation reporter that allows live-imaging of Erk-signalling dynamics. We demonstrate the utility of this line to live-image Erk activity during physiologically relevant angiogenic events. Further, we reveal dynamic and sequential endothelial cell Erk-signalling events following blood vessel wounding. Initial signalling is dependent upon $Ca^{2+}$ in the earliest responding endothelial cells, but is independent of Vegfr-signalling and local inflammation. The sustained regenerative response, however, involves a Vegfr-dependent mechanism that initiates concomitantly with the wound inflammatory response. This work reveals a highly dynamic sequence of signalling events in regenerative angiogenesis and validates a new resource for the study of vascular Erk-signalling in real-time.

## Introduction

The formation of new blood vessels from pre-existing vasculature (angiogenesis) is a fundamental process central in the formation of a viable embryo and in the pathogenesis of many diseases (*Carmeliet and Jain, 2011*; *Chung and Ferrara, 2011*; *Potente et al., 2011*). Angiogenesis is controlled by intricately regulated cell-cell, cell-matrix, and intracellular signalling events. The activity of extracellular signal-regulated kinase (ERK) downstream of the vascular endothelial growth factor A (VEGFA)/VEGF receptor 2 (VEGFR2) signalling pathway is essential for both developmental and

pathological angiogenesis (*Koch and Claesson-Welsh, 2012*; *Simons et al., 2016*). ERK-signalling is also required downstream of VEGFC/VEGFR3-signalling in lymphangiogenesis (*Deng et al., 2013*). ERK is required for angiogenic sprouting, proliferation, and migration, with genetic or pharmacological inhibition of ERK-signalling resulting in impaired blood vessel development in zebrafish and mice (*Srinivasan et al., 2009*; *Costa et al., 2016*; *Nagasawa-Masuda and Terai, 2016*; *Shin et al., 2016a*). Cancer-associated vessels have high ERK activity, and inhibition of ERK-signalling blocks cancer-associated angiogenesis in mice (*Wilhelm et al., 2004*; *Murphy et al., 2006*). Beyond the formation of new vessels, ERK-signalling is also essential to maintain vascular integrity in quiescent endothelial cells (ECs) (*Ricard et al., 2019*), altogether demonstrating a central role for ERK in vascular biology.

Despite its importance, vascular ERK-signalling has largely been examined with biochemical assays or in tissues in static snapshots. Numerous studies have suggested that ERK-signalling is likely to be highly dynamic during angiogenic events; for example, studies that examine Erk activation using antibodies to detect phosphorylated Erk (pErk) have observed changes associated with increased EC signalling, EC motility, and specific EC behaviours (*Costa et al., 2016*; *Nagasawa-Masuda and Terai, 2016*; *Shin et al., 2016a*). In zebrafish, live-imaging of blood ECs at single-cell resolution coupled with carefully staged immunofluorescence staining for pErk suggested that an underlying dynamic Erk-signalling event may control tip-cell maintenance in angiogenesis (*Costa et al., 2016*). Nevertheless, EC-signalling dynamics at the level of key intracellular kinases, such as ERK, remains poorly understood. This gap in our understanding is largely due to a gap in our ability to live-image changes in signalling as they occur.

A number of new biosensors have now been applied in vitro and in vivo that allow live-imaging of proxy readouts for intracellular signalling events (reviewed in detail in *Shu, 2020*). One approach used has involved application of biosensors that utilise fluorescence resonance energy transfer (FRET)-based readouts. The first ERK FRET-based biosensor (ERK activity reporter (EKAR)) was developed in 2008 (*Harvey et al., 2008*). Since then, modifications have been made to improve sensitivity and dynamic range and to generate other ERK FRET-based biosensors such as EKAR-EV, RAB-EKARev, and sREACh (*Komatsu et al., 2011*; *Ding et al., 2015*; *Tang and Yasuda, 2017*; *Mehta et al., 2018*). Importantly, these ERK FRET-based biosensors have been applied in vivo to visualise ERK-signalling dynamics in various cell types during development, cell migration, and wound healing (*Kamioka et al., 2012*; *Mizuno et al., 2014*; *Goto et al., 2015*; *Hiratsuka et al., 2015*; *Kamioka et al., 2017*; *Takeda and Kiyokawa, 2017*; *Sano et al., 2018*; *Wong et al., 2018*). While ERK FRET-based biosensors have been widely reported, they are limited in requiring extensive FRET controls and a low speed of acquisition for FRET-based imaging. More recently, *Regot et al., 2014* generated the ERK-kinase translocation reporter (KTR)-Clover construct (hereafter referred to as EKC), which allows for dynamic analysis of ERK activity using a readout not involving FRET. A fluorescence-based kinase activity reporter translates ERK phosphorylation events into a nucleo-cytoplasmic shuttling event of a synthetic reporter (*Regot et al., 2014*). Thus, the KTR system allows rapid quantifiable measurements of ERK activity based upon subcellular localisation of a fluorescent fusion protein and is more sensitive to phosphatase-mediated kinase activity downregulation when compared to other commonly used reporters. This has been applied to enable dynamic ERK-signalling pulses to be analysed at single-cell resolution both in vitro and in vivo (*Regot et al., 2014*; *de la Cova et al., 2017*; *Mayr et al., 2018*; *Goglia et al., 2020*; *Pokrass et al., 2020*; *De Simone et al., 2021*), where it has been demonstrated to be of high utility.

In this study, we generated a vascular EC-restricted EKC zebrafish transgenic strain and assessed its utility to study angiogenesis in vivo. We applied real-time quantification of Erk-signalling dynamics during developmental angiogenesis and vessel regeneration. We validated methods to quantify Erk activity during real-time imaging that will be applicable in a variety of settings in vascular biology and beyond. Demonstrating the promise of this approach, we here identify an immediate early Erk-signalling response to wounding of vasculature that is $Ca^{2+}$ signalling dependent and distinct from a later Vegfr-driven regenerative response. Overall, this work reports a unique resource for imaging of vascular signalling and further illuminates mechanisms of vascular regeneration following wounding.

## Results

### Generation of a zebrafish EC-EKC transgenic line

KTRs utilise a kinase docking and target site that is juxtaposed to a phospho-inhibited nuclear localization signal (NLS) and attached to a fluorescent tag (*Regot et al., 2014*). Upon kinase activity, the NLS is inactive and the fluorescent tag detected in the cytoplasm, and when the kinase is not active, dephosphorylated NLS leads to increased nuclear localisation. The EKC module that we took advantage of here relies upon the well-characterised ERK-dependent transcription factor ELK1, utilising the ERK docking site (*Figure 1A*; *Chang et al., 2002*; *Regot et al., 2014*). This reporter has previously been shown to report Erk activity in vivo (*de la Cova et al., 2017*; *Mayr et al., 2018*; *Pokrass et al., 2020*; *De Simone et al., 2021*). To visualise real-time Erk-signalling in ECs, we expressed this reporter under the control of an EC-specific promoter (*fli1aep*; *Villefranc et al., 2007*; *Figure 1A–E*). Blood vessel development was unaffected in *Tg(fli1aep:EKC)* transgenic embryos and larvae (*Figure 1B–E*). Furthermore, transgenic adults displayed no adverse morphological features and were fertile (data not shown), indicating that EKC does not inhibit Erk-signalling in vivo, or cause developmental phenotypes, consistent with previous findings (*Mayr et al., 2018*; *De Simone et al., 2021*).

To test if the *Tg(fli1aep:EKC)* line reports vascular Erk-signalling, embryos were treated with either dimethyl sulfoxide (DMSO), mitogen-activated protein kinase kinase (MEK) inhibitor SL327, or pan-VEGFR inhibitor SU5416, and vascular EKC localisation examined at 28 hours post-fertilisation (hpf). Tip ECs in developing intersegmental vessels (ISVs) have been shown to have high Erk activity (*Costa et al., 2016*; *Nagasawa-Masuda and Terai, 2016*; *Shin et al., 2016a*), and we observed nuclear depleted EKC localisation in leading angiogenic ISV cells including at the level of the dorsal longitudinal anastomotic vessel (DLAV) in DMSO-treated embryos (*Figure 1F–F'',I*). In contrast, ISV ECs of embryos treated with either SL327 or SU5416 had nuclear enriched EKC localisation, indicating inactive Erk-signalling (*Figure 1G–I*). To best visualise these differences in signalling and differences shown below, we used a heat map of nuclear EC EKC intensity that is inverted so that blue-scale indicated low signalling (nuclear enriched) and red-scale indicated high signalling (nuclear depleted) (*Figure 1F''–H''*). Therefore, we confirmed that the *Tg(fli1aep:EKC)* (hereafter EC-EKC) transgenic line enables quantification of Erk activity in developing ECs.

### The EC-EKC line enables visualisation and quantification of dynamic Erk activity during primary angiogenesis

We next sought to determine whether the EC-EKC line reports physiologically relevant Erk-signalling events. Using immunofluorescence staining, ISV tip cells that sprout from the dorsal aorta (DA) have been shown to have higher Erk-signalling than ECs that remain in the DA during the initiation of angiogenesis (*Nagasawa-Masuda and Terai, 2016*; *Shin et al., 2016a*). We examined 22 hpf embryos and indeed observed that sprouting ISV ECs display high Erk activity based upon EKC localisation (*Figure 1—figure supplement 1A–B*). However, many DA ECs also appeared to have nuclear depleted EKC localisation (*Figure 1—figure supplement 1A*, yellow arrows). To compare EKC- and Erk-signalling levels between sprouting tip cells and the DA, we utilised multiple methods. We found that measuring the nuclear/cytoplasm EKC intensity ratio in DA ECs was inaccurate because DA ECs are irregular in morphology, making cytoplasmic quantification unreliable (*Figure 1—figure supplement 1A'*). Previous studies have compared nuclear EKC with nuclear H2B-mCherry intensity in the same cell as a ratio to measure Erk activity (e.g., in vulval precursor cells in the worm; *de la Cova et al., 2017*). We assessed the ratio of nuclear EKC/H2B-mCherry intensity in double transgenic EC-EKC;*Tg(fli1a:H2B:mCherry)* embryos and found that the ISV tip cells had higher Erk activity than adjacent DA 'stalk' ECs (*Figure 1—figure supplement 1A'' and C*). We used a stable *Tg(fli1a:H2B-mCherry)* transgenic line with consistent H2B-mCherry intensity. Next, we investigated whether nuclear EKC intensity alone was sufficient to compare Erk-signalling between ECs. The ratio of nuclear EKC intensity of the sprouting ISV tip cell and the adjacent DA 'stalk' EC clearly showed higher signalling in tip cells and was consistent with EKC/H2B-mCherry measurements (*Figure 1—figure supplement 1C*). Thus, we establish that both methods can be reliably used, when measurement of nuclear/cytoplasm EKC intensity is not possible because of difficulty in identifying a cell's cytoplasm. We compare nuclear EKC intensities for analyses hereafter.

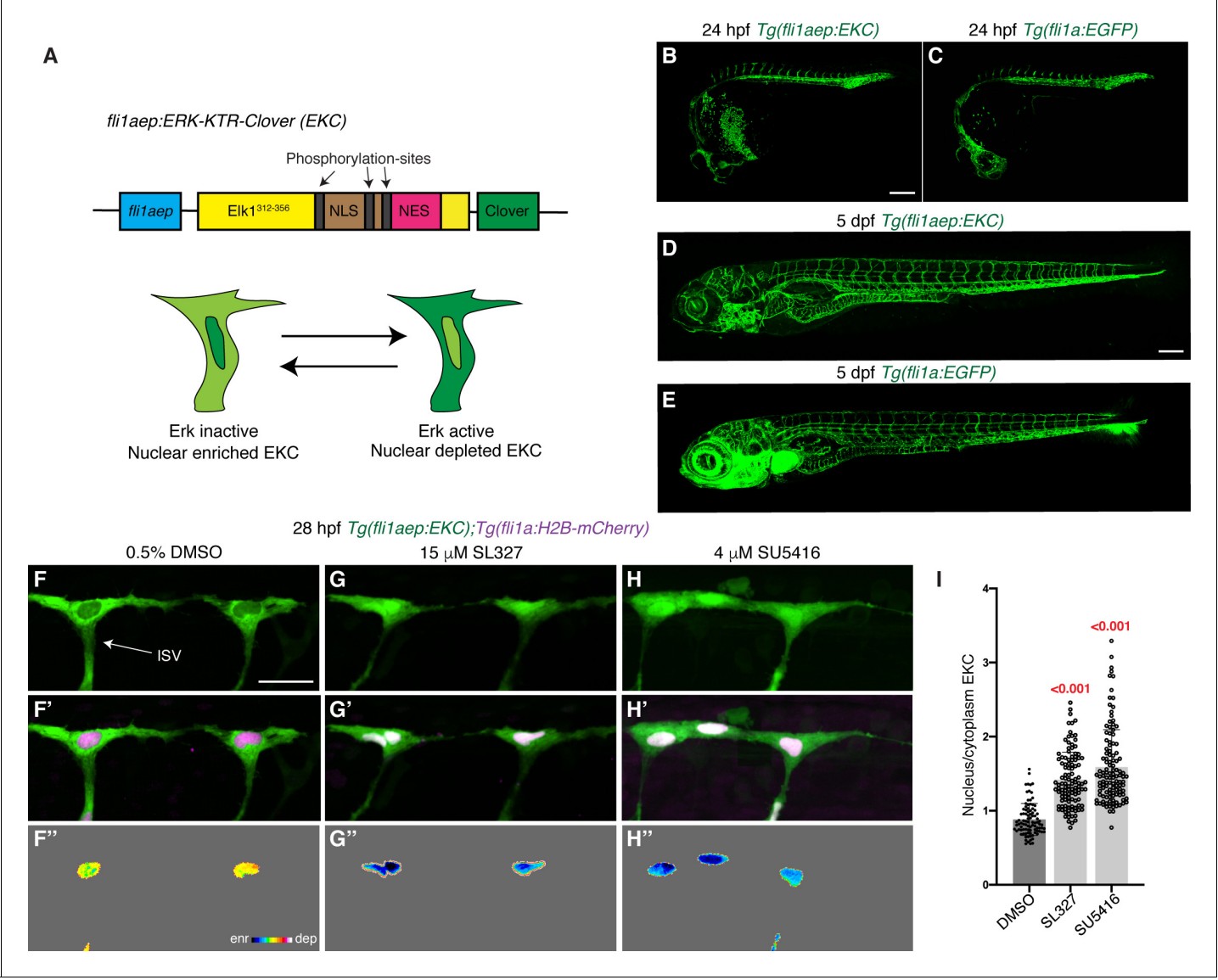

**Figure 1.** The EC-EKC transgenic line enables quantification of vascular Erk activity during development. (**A**) Schematic representation of the *fli1aep: ERK-KTR-Clover* (EKC) construct, and endothelial cells (ECs) with nuclear enriched EKC (bottom left, inactive Erk-signalling) and nuclear depleted EKC localisation (bottom right, active Erk-signalling). (**B–E**) Lateral confocal images of the EC-EKC (**B,D**) and *Tg(fli1a:EGFP)* (**C,E**) embryos/larvae at 24 hours post-fertilisation (hpf) (**B,C**) and 5 days post-fertilisation (dpf) (**D,E**). Blood vessel development is not altered in EC-EKC embryos/larvae. (**F–H''**) Lateral spinning disc confocal images of ISV ECs in 28 hpf EC-EKC embryos treated for 1 hr with either 0.5% dimethyl sulfoxide (DMSO) (**F–F''**), with active EC Erk-signalling, or 15 μM SL327 (**G–G''**) or 4 μM SU5416 (**H–H''**), all of which with inactive EC Erk-signalling. Images (**F-H**) show *fli1aep:EKC* expression, while images (**F'-H'**) show both *fli1aep:EKC* and *fli1a:H2B-mCherry* expression. Images (**F''-H''**) show the nuclear *fli1aep:EKC* expression with intensity difference represented in 16 colour LUT (Fiji). The *fli1a:H2B-mCherry* signal was used to mask the nucleus. (**I**) Quantification of nucleus/ cytoplasm EKC intensity in ISV tip ECs of 28 hpf embryos treated with either 0.5% DMSO (0.881, 93 ISV tip ECs, n = 20 embryos), 15 μM SL327 (1.419, 114 ISV tip ECs, n = 27 embryos), or 4 μM SU5416 (1.591, 118 ISV tip ECs, n = 27 embryos). ISV: intersegmental vessel. Statistical test: Kruskal-Wallis test was conducted for graph (**I**). Error bars represent standard deviation. Scale bars: 200 μm for images (**B**) and (**D**), 25 μm for image (**F**). The online version of this article includes the following source data and figure supplement(s) for figure 1:

**Source data 1.** Nuclear/cytoplasm EKC measurements in leading ISV ECs of DMSO-, SL327-, and SU5416-treated 28 hpf embryos.

**Figure supplement 1.** The EC-EKC transgenic line reports tip-cell-enriched and cell-state-dependent Erk-signalling during primary angiogenesis.

**Figure supplement 1—source data 1.** EKC measurements in ISV ECs at 22 and 28 hpf.

Next, we correlated EC Erk-signalling state (based on EKC intensity) with a cell's migratory state (based on nuclear ellipticity) as previous studies have suggested a correlation (*Costa et al., 2016*). At 28 hpf, ISV tip cells were either located above the horizontal myoseptum with elliptical nuclei indicative of a migrating EC, or at the level of the future DLAV, with less-elliptical (oblate) nuclei indicative of a non-migrating EC (*Figure 1—figure supplement 1D–F*). We found that migrating ECs had higher Erk activity than non-migrating ECs, irrespective of their tip or stalk cell location in an ISV (*Figure 1—figure supplement 1D–G*). This is consistent with previous studies of Vegfa/Kdr/Kdrl/Erk-signalling in zebrafish ISVs (*Yokota et al., 2015*; *Costa et al., 2016*; *Nagasawa-Masuda and Terai, 2016*; *Shin et al., 2016a*) and confirms a strong correlation between ISV EC motility and EC Erk-signalling.

Using carefully staged immunofluorescence analyses, it was previously suggested that when tip cells divide in ISV angiogenesis, daughter cells show asymmetric Kdrl/Erk-signalling that re-establishes the tip/stalk EC hierarchy (*Costa et al., 2016*). However, an analysis of fixed material can only ever imply the underlying dynamics. To investigate the dynamics of Erk-signalling upon tip-cell division, we performed high-speed time-lapse imaging of ISV tip ECs as they undergo mitosis in 24 hpf embryos. Immediately preceding cell division, ECs displayed cytoplasmic localisation of H2B-mCherry due to the disruption of the nuclear membrane (*Figure 2A*, yellow arrow). High-speed live-imaging of ISV tip ECs revealed nuclear enriched EKC localisation during cell division (*Figure 2A–C*), which was maintained until cytokinesis (*Figure 2B*, *Video 1*) but may reflect nuclear membrane breakdown rather than altered cellular signalling. Subsequently, daughter ECs in the tip position progressively increased their Erk activity, while ECs in the trailing stalk daughter position remained nuclear enriched, showing asymmetric Erk-signalling activity rapidly following cell division (*Figure 2B–I*, *Video 1*). To accurately assess this across multiple independent tip-cell divisions, we measured the ratio of tip/stalk daughter cell nuclear EKC intensity over time. This revealed that tip cells consistently increased their Erk activity relative to stalk cells in a progressive manner with the most dramatic asymmetry observed ~21 min post-cytokinesis (*Figure 2J,K*, *Video 1*). Collectively, the EC-EKC line enabled quantitative assessment of physiologically relevant Erk activity by real-time live-imaging and confirmed previously suggested asymmetric signalling post tip-cell division.

## Vessel wounding induces rapid Erk activation

As a major downstream target for VEGFA/VEGFR2 signalling, ERK is essential for stimulating ectopic sprouting of otherwise quiescent mature vessels (*Wilhelm et al., 2004*; *Murphy et al., 2006*). However, Erk-signalling dynamics during pathological angiogenesis has not been analysed in detail. To determine whether the EC-EKC line can be used to dynamically visualise Erk activation in ECs in pathological settings, we analysed EC Erk activity following vessel wounding using a laser ablation method. We chose this model because vessel wounding in 4 days post-fertilisation (dpf) larvae results in highly reproducible Vegfa/Kdr/Kdrl signalling-dependent vessel regeneration (*Gurevich et al., 2018*). Importantly, cell wounding induces ERK-signalling in vitro and in vivo in other settings (*Matsubayashi et al., 2004*; *Li et al., 2013*; *Hiratsuka et al., 2015*; *Aoki et al., 2017*; *Mayr et al., 2018*).

To visualise Erk-signalling dynamics following cellular ablation and vessel wounding, we time-lapse imaged both ablated ISV ECs and the adjacent non-ablated ISV ECs in 4 dpf EC-EKC larvae for 20 min before and for 22 min after vessel wounding (*Figure 3A–C*). As a control, ISV ECs of unablated 4 dpf larvae were time-lapse imaged for the same period. EKC localisation in the majority of ISV ECs indicated low basal Erk-signalling in ECs of mature vessels (*Figure 3D,D',F,F',H,I*, *Videos 2–5*). Upon vessel wounding, Erk activity in ablated ISV ECs immediately increased (*Figure 3E,E',H,I*, *Videos 3* and *4*). Surprisingly, Erk activity in ECs of ISVs located adjacent to the ablated ISV (termed adjacent ISV) also rapidly increased (*Figure 3G,G',H,I*, *Videos 3* and *5*). Although the activation of Erk-signalling in adjacent ISV ECs was slower than that in ablated ISV ECs, the level of Erk activation in ablated and adjacent vessels was comparable by 15 min post-ablation (mpa, green dotted line) and consistent up to 22 mpa (*Figure 3I*). Both venous and arterial ECs equally showed Erk activation 15 mpa in ablated ISVs post-vessel wounding, suggesting that both venous and arterial ECs are able to rapidly activate Erk-signalling (*Figure 3J*). Finally, to understand the relationship between Erk activation in vessels and distance from the wound, we measured the response of ECs in immediately adjacent, second adjacent, and third adjacent ISVs from the ablated ISV (in an anterior direction).

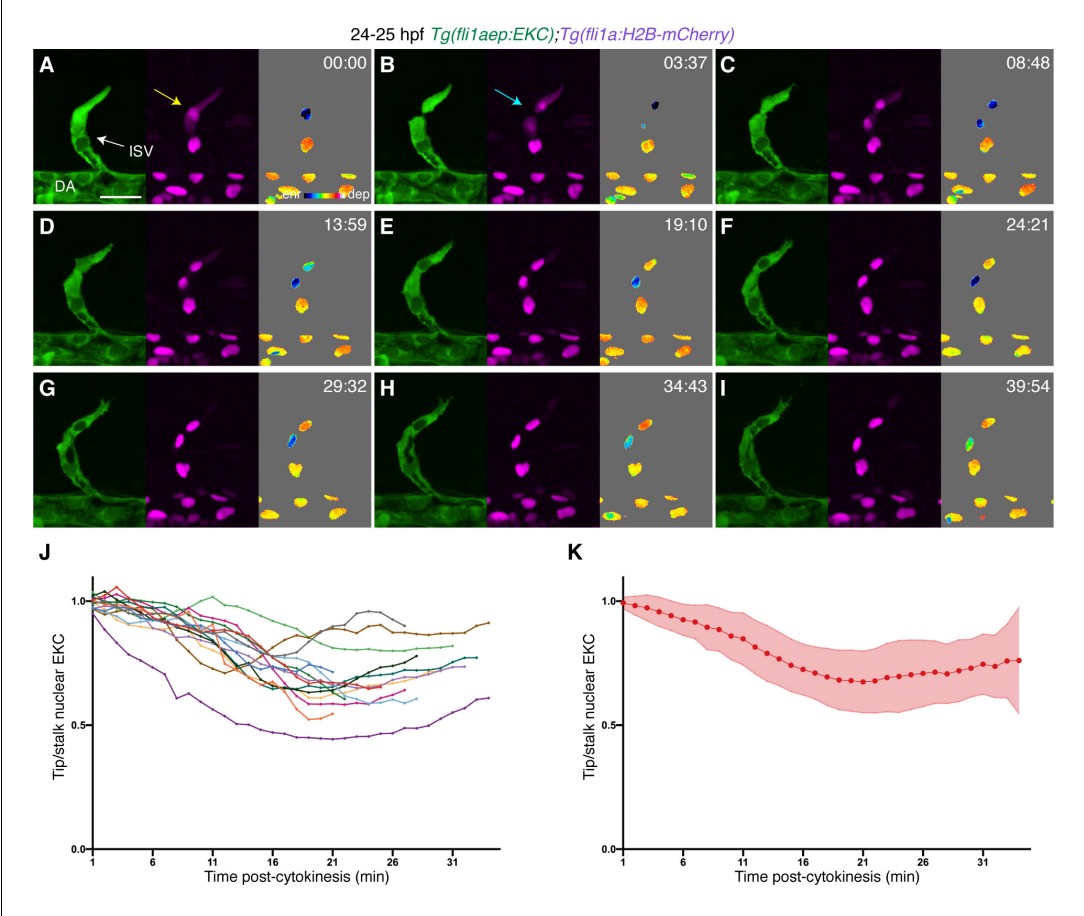

**Figure 2.** Tip cells show asymmetric Erk activity immediately following cell division. (A–I) Still images from *Video 1* showing ISV endothelial cells (ECs) in a 24–25 hours post-fertilisation (hpf) EC-EKC embryo at indicated timepoints. The tip daughter cell has higher Erk activity when compared to the stalk daughter cell immediately after cell division. Left panels show *fli1aep:EKC* expression, middle panels show *fli1a:H2B-mCherry* expression, and right panels show the nuclear *fli1aep:EKC* intensity. The *fli1a:H2B-mCherry* signal was used to mask the nucleus. The yellow arrow indicates a tip ISV EC with cytoplasmic *fli1a:H2B-mCherry* expression. The light blue arrow indicates a tip ISV EC that has undergone cytokinesis. (J,K) Quantification of tip/stalk nuclear EKC intensity of daughter ECs post-cytokinesis (14 EC division events, n = 14 embryos). Graph (J) shows quantification of individual biological replicates and graph (K) shows the average of all biological replicates. ISV: intersegmental vessel; DA: dorsal aorta. Error bars represent standard deviation. Scale bar: 25 μm.

The online version of this article includes the following source data for figure 2:

**Source data 1.** Tip/stalk nuclear EKC measurements in ISV ECs following cell division.

We found that the activation of Erk-signalling was largely limited to the wounded and immediately adjacent ISVs (*Figure 3—figure supplement 1*).

## The initial rapid Erk-signalling response is not induced by macrophages or Vegfr activity

Macrophages recruited to a wound site have been shown to provide a local source of Vegfa that stimulates vessel regeneration (*Gurevich et al., 2018*). Therefore, we investigated whether macrophages are required for rapid Erk activation in ISV ECs. As previously reported (*Gurevich et al., 2018*), macrophage recruitment to the wound was minimal at 15 mpa, while robust macrophage recruitment was observed 3 hr post-ablation (hpa), suggesting that macrophages may not contribute to rapid Erk activation (*Figure 3—figure supplement 2A–D*). We depleted macrophages by knocking down Spi-1 proto-oncogene b (Spi1b) and Colony stimulating factor three receptor (Csf3r) using established morpholino oligomers (*Rhodes et al., 2005*; *Ellett et al., 2011*; *Pase et al., 2012*; *Figure 3—figure supplement 2E–G*). We found that depletion of macrophages led to a quantifiable but mild reduction in normal vessel regeneration measured at 24 hpa in this model (*Figure 3—*

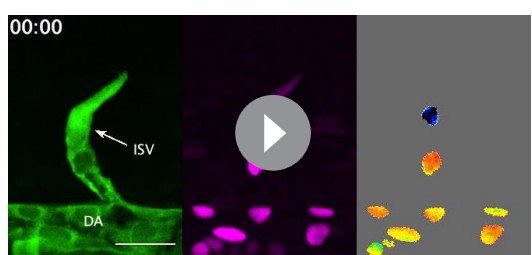

**Video 1.** ISV daughter ECs show asymmetric Erk activity following cytokinesis. Time-lapse video of an ISV tip endothelial cell (EC) undergoing mitosis in a 24–25 hours post-fertilisation (hpf) EC-EKC embryo. Left panel shows the *fli1aep:EKC* expression, middle panel shows the *fli1a:H2B-mCherry* expression, and the right panel shows the nuclear *fli1aep:EKC* intensity. Z stacks were acquired every 15.5 s for 40 min using an Andor Dragonfly Spinning Disc Confocal microscope. Photobleaching was minimised using the bleach correction tool (correction method: histogram matching) in FIJI.    ISV: intersegmental vessel; DA: dorsal aorta. Scale bar: 25 μm.

https://elifesciences.org/articles/62196#video1

*figure supplement 2H–J*). The rapid EC Erk activation post-wounding was unaffected upon macrophage depletion (*Figure 3K*, *Figure 3—figure supplement 2K–V'*). We next tested whether Vegfr-signalling was required for this rapid Erk activation. Erk activation in both ablated and adjacent ISV ECs 15 mpa was blocked in larvae treated with SL327, indicating that it requires upstream Mek activation (*Figure 3L*, *Figure 3—figure supplement 3D–M'*). However, treatment with two independent and validated VEGFR inhibitors, SU5416 (*Figure 1H–I*) and AV951 (*Figure 3—figure supplement 3A–C*), did not impair the rapid Erk activation at 15 mpa (*Figure 3L*, *Figure 3—figure supplement 3O–Z'*). Therefore, at 15 mpa, Erk activation in both ablated and adjacent ISV ECs is induced independently of either macrophages or Vegfr-signalling, suggesting an initial response to vessel wounding that has not been previously examined.

## Following the initial rapid Erk activation, signalling is progressively restricted to regenerating vessels

Previous studies have shown that local wounding induces a rapid burst in ERK-signalling in surrounding cells (*Matsubayashi et al., 2004*; *Li et al., 2013*; *Hiratsuka et al., 2015*; *Aoki et al., 2017*; *Mayr et al., 2018*). To determine whether the initial Erk activation in ISV ECs post-vessel wounding was maintained, Erk activity was followed over a longer time-course until 3 hpa, when robust macrophage recruitment was observed (*Figure 3—figure supplement 2C,D*). Erk activity was again increased upon vessel wounding in both ablated and adjacent ISV ECs at 15 mpa (*Figure 4A–D*, *Figure 4—figure supplement 1A–I'*). Erk activity was maintained until 30 mpa in adjacent ISV ECs, but then gradually decreased and returned to non-ablated control levels by 1 hpa (*Figure 4B–D*). By contrast, high Erk activity was maintained for the duration in ablated ISV ECs (*Figure 4A,A',C,D*). To test if this difference in Erk activity was influenced by long-term time-lapse imaging, Erk-signalling was analysed in ISV ECs of 3 hpa larvae. Similar to the time-course analysis, Erk activity in ablated ISV ECs was high at 3 hpa, while ECs in adjacent ISVs were at non-ablated control level (*Figure 4—figure supplement 1J–N*).

Given that the initial rapid burst of Erk activation progressively returns to basal levels in unwounded vessels, we assessed if this was a general wound response. We examined the initial Erk-signalling burst in muscle and skin cells following a large puncture wound using a ubiquitous EKC strain (*Mayr et al., 2018*). This confirmed that an initial activation of Erk-signalling in cells surrounding the puncture wound was only transient (*Video 6*) and, in this case, was progressively lost even in cells at the immediate site of the wound, unlike in regenerating vessels. To further investigate whether only regenerating ISVs maintain high Erk activity after wounding, tissue in between the ISVs was ablated without injuring the ISVs in 4 dpf EC-EKC larvae (termed control ablation hereafter). Erk activity in surrounding ISV ECs was analysed at 15 mpa and 3 hpa. Similar to vessel ablation, this adjacent tissue ablation resulted in rapid activation of Erk-signalling in ISV ECs (*Figure 4—figure supplement 2A–C*). Erk activity in these ECs decreased to non-ablated control levels by 3 hpa (*Figure 4—figure supplement 2A–C*). Therefore, Erk-signalling is immediately activated in muscle, skin epithelial and ECs upon injury, but only regenerating vessels retain this high activity at 3 hpa upon vessel wounding.

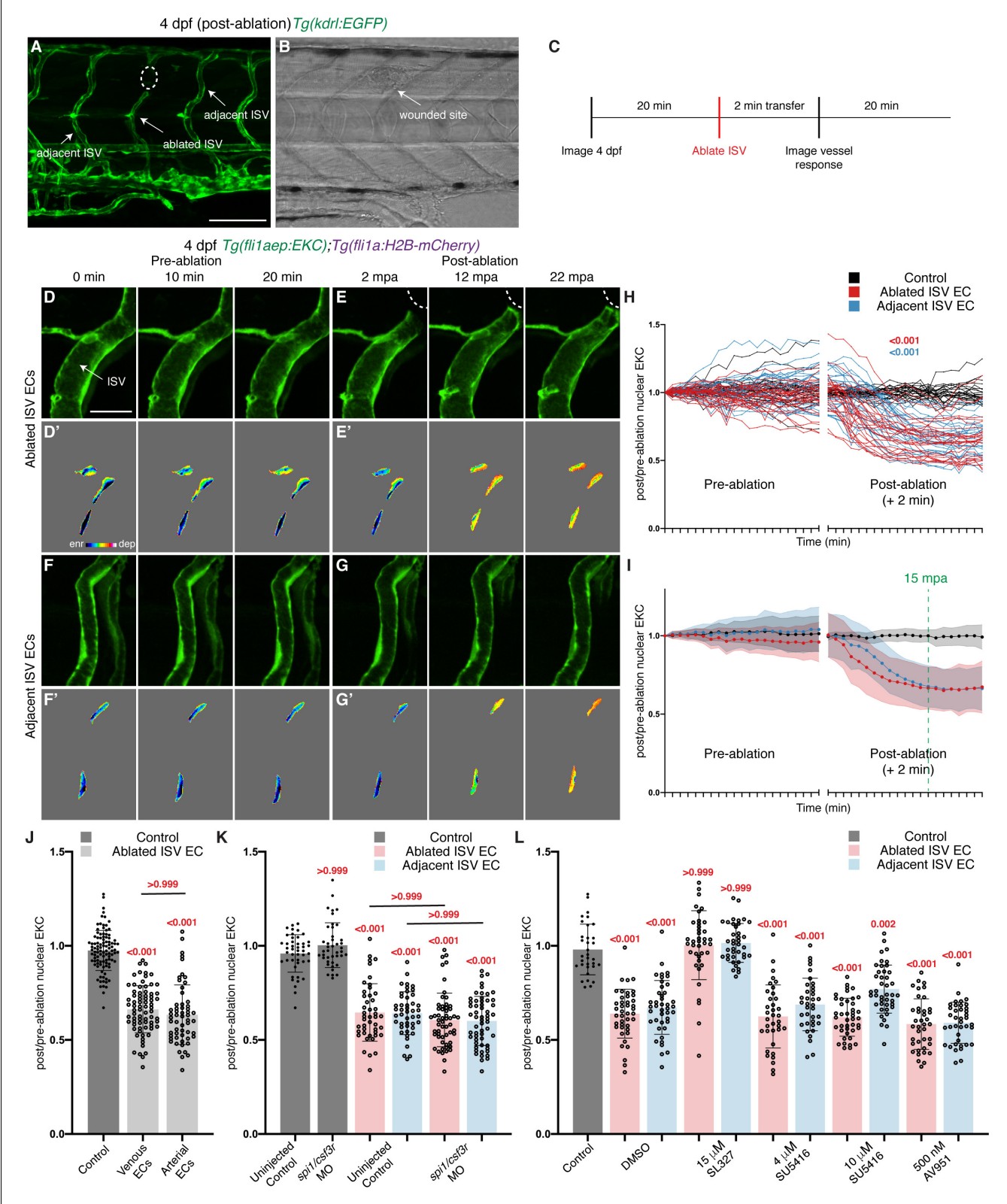

**Figure 3.** Wounded vessels rapidly activate Erk independent of macrophages or Vegfr-signalling. (**A,B**) Lateral confocal images of a 4 days post-fertilisation (dpf) *Tg(kdrl:EGFP)* larva following vessel wounding (post-ablation). Image (**A**) shows the *kdrl:EGFP* expression and image (**B**) shows the trans-light image of image (**A**). Ablated ISV, adjacent ISVs, and the wounded site are indicated with white arrows. (**C**) Schematic representation of imaging schedule for larvae in images (**D-G**) and *Videos 3–5*. (**D–G'**) Still images from *Video 4* (**D–E'**) and *Video 5* (**F–G'**) showing ISV endothelial cells

*Figure 3 continued on next page*

*Figure 3 continued*

(ECs) before (pre-ablation) and after vessel wounding. Ablated and adjacent ISV ECs rapidly activate Erk-signalling. (D-G) *fli1aep:EKC* expression; (D'-G') nuclear intensity. (H,I) Quantification of post-/pre-ablation nuclear EKC intensity of ECs in non-ablated control ISVs (black, 24 ECs, n = 8 larvae), ablated ISVs (red, 27 ECs, n = 9 larvae), and adjacent ISVs (light blue, 27 ECs, n = 9 larvae). (H) shows quantification of individual ECs and (I) shows the average of all biological replicates. Green dotted line indicates 15 min post-ablation (mpa). (J) Quantification of post-/pre-ablation nuclear EKC intensity 15 mpa in ECs of non-ablated control ISVs (103 ECs, n = 34 larvae), ablated venous ISVs (75 ECs, n = 25 larvae), and ablated arterial ISVs (57 ECs, n = 19 larvae). Both venous and arterial ISV ECs activate Erk-signalling. (K) Quantification of post-/pre-ablation nuclear EKC intensity 15 mpa in ECs of non-ablated uninjected control ISVs (45 ECs, n = 15 larvae), non-ablated *spi1/csf3r* morphant ISVs (42 ECs, n = 14 larvae), uninjected control ISVs (45 ablated/adjacent ISV ECs, n = 15 larvae), and *spi1/csf3r* morphant ISVs (56 ablated ISV ECs and 57 adjacent ISV ECs, n = 19 larvae). Macrophages are not required to rapidly activate Erk-signalling in ablated or adjacent ISV ECs. (L) Quantification of post-/pre-ablation nuclear EKC intensity 15 mpa in ECs of 0.5% dimethyl sulfoxide (DMSO)-treated non-ablated control ISVs (33 ECs, n = 11 larvae) and ISVs of larvae treated with either 0.5% DMSO (42 ablated/adjacent ISV ECs, n = 14 larvae), 15 µM SL327 (39 ablated/adjacent ISV ECs, n = 13 larvae), 4 µM SU5416 (36 ablated/adjacent ISV ECs, n = 12 larvae), 10 µM SU5416 (42 ablated/adjacent ISV ECs, n = 14 larvae), or 500 nM AV951 (42 ablated/adjacent ISV ECs, n = 14 larvae). Vegfr-signalling is not required to rapidly activate Erk-signalling in ablated or adjacent ISV ECs. ISV: intersegmental vessel. Statistics: permutation test was conducted for graph (H). Kruskal-Wallis test was conducted for graphs (J-L). Error bars represent standard deviation. White dotted lines/circle shows the wounded sites of each larva. Scale bar: 100 µm for image (A), 20 µm for image (D).

The online version of this article includes the following source data and figure supplement(s) for figure 3:

**Source data 1.** Post-/pre-ablation nuclear EKC measurements in control, ablated, and adjacent ISV ECs.
**Figure supplement 1.** Rapid Erk activation is largely restricted to wounded and adjacent ISV ECs.
**Figure supplement 1—source data 1.** Post-/pre-ablation nuclear EKC measurements in adjacent, second adjacent, and third adjacent ISV ECs 15 mpa.
**Figure supplement 2.** Macrophages are not required for rapid Erk activation following vessel wounding.
**Figure supplement 2—source data 1.** Measurements of macrophage number and ISV length.
**Figure supplement 3.** Vegfr-signalling is not required for rapid Erk activation following vessel wounding.
**Figure supplement 3—source data 1.** Nuclear/cytoplasm EKC measurements in leading ISV ECs of DMSO- and AV951-treated 28 hpf embryos.

## Vegfr-signalling drives ongoing Erk activity to control vessel regeneration

We next examined if ongoing Erk activity in ablated ISV ECs was maintained by Vegfr-signalling consistent with earlier reports (*Gurevich et al., 2018*). To test this, we analysed Erk activity of ablated ISV ECs in 3 hpa larvae treated with inhibitors of the Kdr/Kdrl/Mek/Erk-signalling pathway. Treatment with SL327 inhibited Erk activation at 3 hpa, as did treatment with the Vegfr inhibitor SU5416 (*Figure 5A*, *Figure 5—figure supplement 1A–F',I–J'*). Furthermore, we used an F0 CRISPR approach (*Wu et al., 2018*) to generate *kdrl* knockout embryos (termed *kdrl* crispant hereafter). These embryos phenocopied earlier reported mutant and morphant phenotypes (*Figure 5—figure supplement 1K,L*; *Habeck et al., 2002*; *Covassin et al., 2006*). 3 hpa F0 crispant larvae displayed reduced Erk activity in EC-EKC measurements compared with ISV ablation control larvae (*Figure 5B*, *Figure 5—figure supplement 1M–T'*). Unlike drug-treated larvae, *kdrl* crispants displayed a mild reduction in Erk activity, likely due to compensation from other Vegfrs, such as Kdr, and/or Flt4 (zebrafish orthologue of VEGFR3) (*Covassin et al., 2006*; *Shin et al., 2016b*). Overall, these genetic and pharmacological approaches indicate that Vegfr-/Mek-signalling is required for sustained high Erk activity

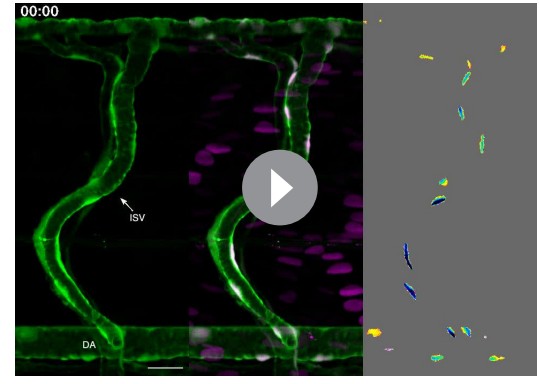

**Video 2.** ISV ECs in 4 dpf larvae have minimal Erk activity. Time-lapse video of the trunk vessels in a 4 days post-fertilisation (dpf) EC-EKC larva at indicated timepoints. Endothelial cells (ECs) in functional vessels at 4 dpf have low Erk activity. Left panel shows the *fli1aep:EKC* expression, middle panel shows both *fli1aep:EKC* and *fli1a:H2B-mCherry* expression, and the right panel shows the nuclear *fli1aep:EKC* intensity. Z stacks were acquired every minute for 41 min using an Andor Dragonfly Spinning Disc Confocal microscope. Photobleaching was minimised using the bleach correction tool (correction method: histogram matching) in FIJI. ISV: intersegmental vessel; DA: dorsal aorta. Scale bar: 20 µm.
https://elifesciences.org/articles/62196#video2

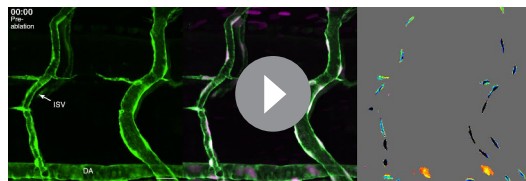

**Video 3.** Both ablated and adjacent ISV ECs rapidly activate Erk-signalling following vessel wounding. Time-lapse video of the trunk vessels in a 4 days post-fertilisation (dpf) EC-EKC larva before (pre-ablation) and after (post-ablation) vessel wounding at indicated timepoints. Vessel wounding rapidly activates Erk-signalling in both ablated and adjacent ISV endothelial cells (ECs). Post-ablation video starts at 2 min post-ablation due to the time taken to transfer the larvae between microscopes and for preparation of imaging. Left panel shows the *fli1aep:EKC* expression, middle panel shows both *fli1aep:EKC* and *fli1a:H2B-mCherry* expression, and the right panel shows the nuclear *fli1aep:EKC* intensity. Z stacks were acquired every 1 min for 20 min before and after vessel wounding using an Andor Dragonfly Spinning Disc Confocal microscope. Photobleaching was minimised using the bleach correction tool (correction method: histogram matching) in FIJI.      ISV: intersegmental vessel; DA: dorsal aorta. Scale bar: 20 μm.
https://elifesciences.org/articles/62196#video3

**Video 4.** Ablated ISV ECs rapidly activate Erk-signalling following vessel wounding. Time-lapse video of the ablated ISV in a 4 days post-fertilisation (dpf) EC-EKC larva before (pre-ablation) and after (post-ablation) vessel wounding at indicated timepoints. Post-ablation video starts at 2 min post-ablation due to the time taken to transfer the larvae between microscopes and for preparation of imaging. Left panel shows the *fli1aep:EKC* expression and the right panel shows the nuclear *fli1aep:EKC* intensity. Z stacks were acquired every 1 min for 20 min before and after vessel wounding using an Andor Dragonfly Spinning Disc Confocal microscope. Photobleaching was minimised using the bleach correction tool (correction method: histogram matching) in FIJI.      ISV: intersegmental vessel. Scale bar: 20 μm.
https://elifesciences.org/articles/62196#video4

in ablated ISV ECs at 3 hpa. To determine the functional relevance of this in ongoing regeneration, we treated embryos following ablation-based wounding with SU5416 or two independent Mek inhibitors, SL327 and Trametinib. We observed that inhibition of Vegfr- or Erk-signalling completely blocked all ongoing vessel regeneration (*Figure 5C*, *Figure 5—figure supplement 1U–X*). Finally, we found no difference in EC-EKC activation at 3 hpa in the absence of macrophages, suggesting that macrophages play a modulatory role in vessel regeneration and are not the sole source of Vegfs in this laser ablation model (*Figure 5D*, *Figure 5—figure supplement 2*).

Interestingly, we noted that while treatment with SU5416 at 10 μM blocked ongoing Erk activation (*Figure 5A*, *Figure 5—figure supplement 1I–J'*), treatment with the same inhibitor at a lower dose of 4 μM did not completely block Erk activity (*Figure 5A*, *Figure 5—figure supplement 1G–H'*). To further investigate this with more spatial resolution, we examined Erk activity in ISV ECs relative to their distance from the cellular ablation site. Erk-signalling in the first, second, and third ISV ECs from the wound was activated 3 hpa in control larvae, while treatment with 10 μM SU5416 inhibited signalling in ECs located in all of these positions (*Figure 5E,G,H*, *Figure 5—figure supplement 1C–D',I–J'*). However, with the intermediate dose of 4 μM SU5416, while the closest cell to the wound site still displayed Erk activity, as did the second cell from the wound site, the third and farthest cells from the wounded sites were now inhibited (*Figure 5F,H*, *Figure 5—figure supplement 1G–H'*). These results suggest that there is a gradient of Vegfr/Erk-signalling activity in the ablated ISV ECs resulting in higher Vegfr/Erk activity in ECs closer to the wounded site, which can only be inhibited with SU5416 at higher concentrations. To test this, we examined the EC-EKC levels relative to cell position and directly confirmed this graded activation at 3 hpa (*Figure 5I*, *Figure 4—figure supplement 1J–M'*). Together, these analyses confirm that during the ongoing response to vessel wounding, Vegfr-signalling is crucial and drives a positionally graded signalling response to regulate regenerating vessels.

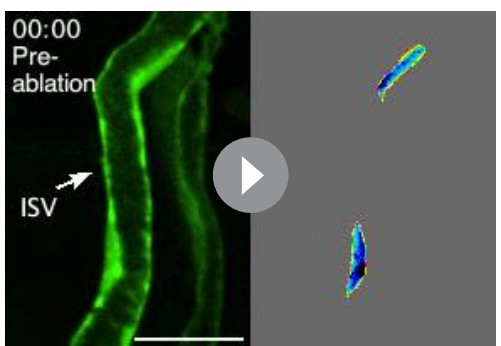

**Video 5.** Adjacent ISV ECs rapidly activate Erk-signalling following vessel wounding. Time-lapse video of the adjacent ISV in a 4 days post-fertilisation (dpf) EC-EKC larva before (pre-ablation) and after (post-ablation) vessel wounding at indicated timepoints. Post-ablation video starts at 2 min post-ablation due to the time taken to transfer the larvae between microscopes and for preparation of imaging. Left panel shows the *fli1aep:EKC* expression and the right panel shows the nuclear *fli1aep:EKC* intensity. Z stacks were acquired every 1 min for 20 min before and after vessel wounding using an Andor Dragonfly Spinning Disc Confocal microscope. Photobleaching was minimised using the bleach correction tool (correction method: histogram matching) in FIJI. ISV: intersegmental vessel. Scale bar: 20 μm.

https://elifesciences.org/articles/62196#video5

## Ca²⁺ signalling is required for initial rapid Erk activation upon vessel wounding

Although Vegfr-signalling is required for sustaining high Erk activity in ablated ISV ECs, it is not required for inducing the initial rapid Erk-signalling response. Activated by ATP released by damaged cells, Ca$^{2+}$ signalling is one of the first intra-cellular mechanisms to be activated post-wounding in many cell types (reviewed in detail in *Ghilardi et al., 2020*). Accordingly, mechanical injury of blood vessels has been shown in situ to rapidly activate Ca$^{2+}$ signalling in neighbouring endothelial cells in excised rat aorta (*Berra-Romani et al., 2008*; *Berra-Romani et al., 2012*). Although Ca$^{2+}$ signalling activates Erk-signalling in endothelial cells downstream of the Vegfa/Vegfr2 signalling pathway (*Koch and Claesson-Welsh, 2012*; *Moccia et al., 2012*), Ca$^{2+}$ signalling alone can also activate Erk-signalling (*Xiao et al., 2011*; *Handly et al., 2015*).

To determine whether Ca$^{2+}$ signalling is rapidly activated in ablated ISV ECs in our model, we measured the dynamic expression of a ubiquitously expressed GCamp, a GFP-based Ca$^{2+}$ probe, using the *Tg(actb2:GCaMP6f);Tg(kdrl:mCherry-CAAX)* transgenic line (*Herzog et al., 2019*). We used a validated transgenic line, which has previously demonstrated a general Ca$^{2+}$ wound response and Ca$^{2+}$ signalling in brain tumours and associated microglia (*Chia et al., 2019*; *Herzog et al., 2019*). We observed a general response in tissue surrounding the ablated site (data not shown), as well as active Ca$^{2+}$ signalling in immune cells (*Figure 6A*, *Videos 7* and *8*, as previously described in *Yoo et al., 2012*; *Razzell et al., 2013*; *de Oliveira et al., 2014*; *Beerman et al., 2015*; *Herzog et al., 2019*; *Poplimont et al., 2020*) in the same movies analysed below, validating the utility of this line. ISVs in non-ablated 4 dpf larvae did not show Ca$^{2+}$ signalling, indicating low Ca$^{2+}$ activity in stable ISVs (*Figure 6B*, *Video 7*). In contrast, ablated ISV ECs showed a rapid pulse of active Ca$^{2+}$ signalling at 5 mpa, which progressively decreased and returned to the level of the surrounding tissue (*Figure 6A,B*, *Video 8*). Active Ca$^{2+}$ signalling was not observed in adjacent ISVs (*Figure 6A,B*, *Video 8*). To determine whether Ca$^{2+}$ signalling is required for rapid Erk activation in ablated ISV ECs, 4 dpf EC-EKC larvae were treated with either DMSO or a potent Ca$^{2+}$ signalling inhibitor Nifedipine for 30 min. Nifedipine treatment did not inhibit Erk-signalling activation in adjacent ISV ECs resulting in similar Erk activity as DMSO-treated larvae 15 mpa (*Figure 6C*, *Figure 6—figure supplement 1A–B',G–J'*). However, Erk activation in ablated ISV ECs (where we observed the GCaMP signal above) was significantly reduced when compared to DMSO-treated larvae (*Figure 6C*, *Figure 6—figure supplement 1C–F'*). This was reproduced in an independent experiment using Amlodipine, an alternative Ca$^{2+}$ signalling inhibitor (*Figure 6D*, *Figure 6—figure supplement 1K–T'*). This indicates not only that Ca$^{2+}$ signalling plays a crucial role upstream of Erk in the wound response, but also that the response is differentially regulated in ablated compared with adjacent vessels, indicative of additional underlying signalling complexity.

We next tested whether Ca$^{2+}$ signalling is required for maintaining Erk activity in ablated ISV ECs 3 hpa. To assess ongoing signalling, 4 dpf EC-EKC larvae were treated with either DMSO or Nifedipine 30 min prior to the 3 hpa timepoint. Activation of Erk-signalling in ablated ISV ECs 3 hpa was not inhibited by Nifedipine (*Figure 6E*, *Figure 6—figure supplement 2A–G'*). Inhibition of Ca$^{2+}$ signalling immediately following wounding between 0 and 30 mpa also had no impact on later Erk-

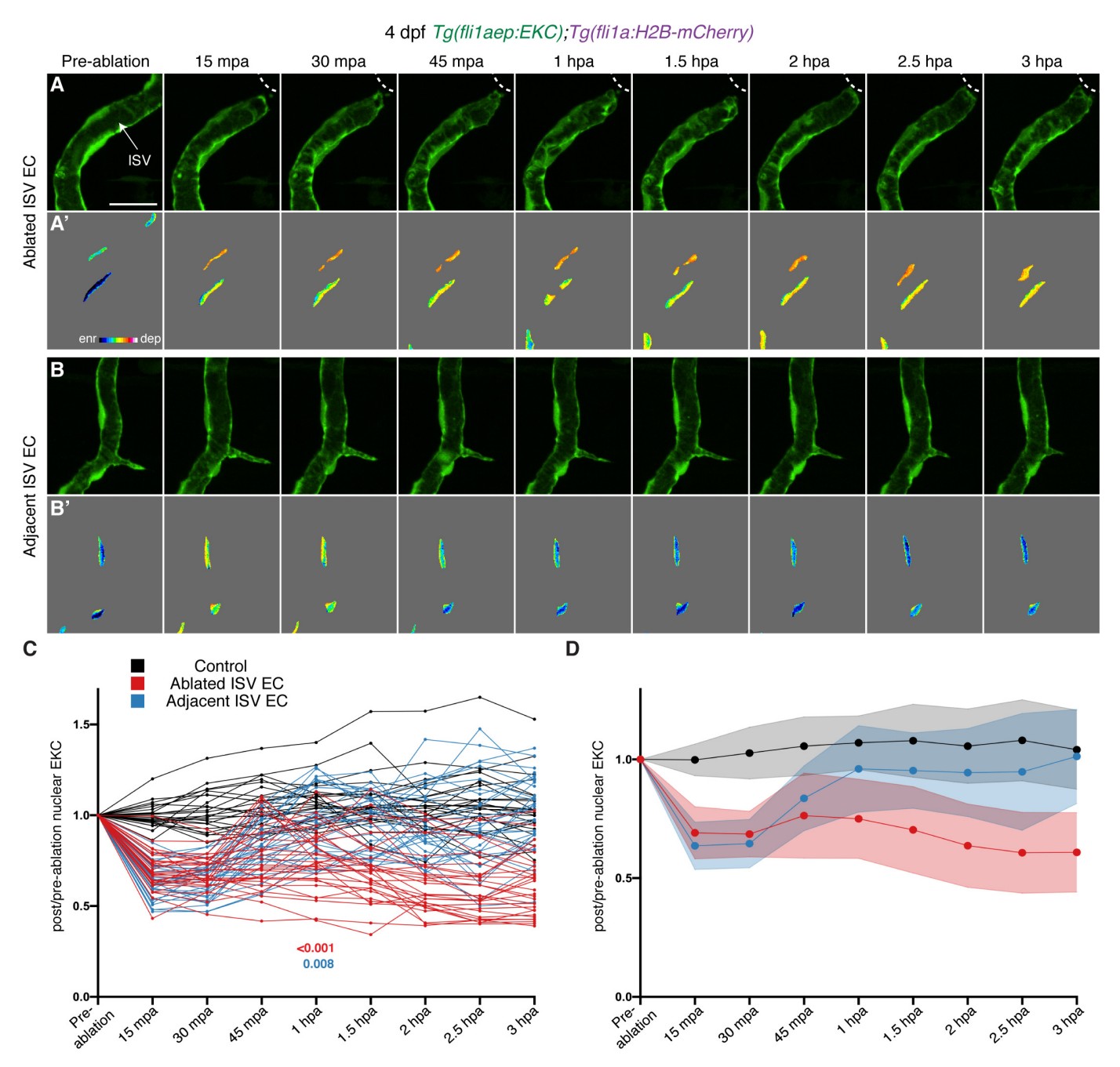

**Figure 4.** Wounded but not adjacent vessels maintain high Erk activity as the regenerative response proceeds. (**A–B'**) Lateral spinning disc confocal images of ablated (**A**) and adjacent ISVs (**B**) of a 4 days post-fertilisation (dpf) EC-EKC larva before and following vessel wounding at indicated timepoints. Erk activity is progressively lost in the adjacent but retained in the wounded ISV endothelial cells (ECs). Images (**A**) and (**B**) show *fli1aep:EKC* expression, and images (**A'**) and (**B'**) show nuclear *fli1aep:EKC* intensity. White dotted lines show the wounded site. (**C,D**) Quantification of post-/pre-ablation nuclear EKC intensity of ECs in non-ablated control ISVs (black, 24 ECs, n = 8 larvae), ablated ISVs (red, 30 ECs, n = 10 larvae), and adjacent ISVs (light blue, 30 ECs, n = 10 larvae) before and after vessel wounding at indicated timepoints. Graph (**C**) shows the quantification of individual ECs and graph (**D**) shows the average of all biological replicates. At 1 hour post-ablation (hpa): control vs ablated ISV ECs: p>0.001; control vs adjacent ISV ECs: p=0.108 (Kruskal-Wallis test). ISV: intersegmental vessel. Statistical test: permutation test was conducted for graph (**C**). Error bars represent standard deviation. Scale bar: 20 μm.

The online version of this article includes the following source data and figure supplement(s) for figure 4:

**Source data 1.** Post-/pre-ablation nuclear EKC measurements in control, ablated, and adjacent ISV ECs from pre-ablation to 3 hpa.

**Figure supplement 1.** Distinct Erk activity between ablated and adjacent ISV ECs 3 hpa.

*Figure 4 continued on next page*

*Figure 4 continued*

**Figure supplement 1—source data 1.** Post-/pre-ablation nuclear EKC measurements in control, ablated, and adjacent ISV ECs at 3 hpa.
**Figure supplement 1—source data 2.** Post-/pre-ablation nuclear EKC measurements in control and control ablated ISV ECs at 15 mpa and 3 hpa.
**Figure supplement 2.** Vessel wounding is required for sustained Erk activity in ablated ISV ECs.

signalling at 3 hpa (*Figure 6F*, *Figure 6—figure supplement 2H–N'*). Thus, Ca$^{2+}$ signalling is required for rapid Erk activation, but not for maintaining Erk activity in ablated ISV ECs. In the analysis of Ca$^{2+}$ signalling following vessel wounding, we noted that this transient pulse of Ca$^{2+}$ signalling was highest in the ECs closest to the wounded site (*Video 8*). Thus, we further sought to determine if Erk-signalling in ECs closest to the wound activates first during the initial dynamic induction. Quantitative analysis based on multiple movies (including *Video 3*) showed that Erk-signalling in ECs proximal to the wounded site (first and second positioned ECs) activated first, followed by ECs farther away from the wounded site (third, fourth, and fifth ECs) (*Figure 6G*). Quantitatively, the ECs proximal to the ablation site (first and second positioned ECs) showed the highest magnitude of difference from control, and this difference reduced as ECs were positioned farther from the ablation site (*Figure 6H*). This shows that like the initial burst in Ca$^{2+}$ signalling post-vessel wounding, Erk-signalling is activated progressively in ECs closest to the wounded site first, followed by those farther away.

## Discussion

ERK-signalling is a downstream target for a number of pathways essential for development (including VEGFA/VEGFR2, EGF/EGFR, and FGF/FGFR pathways) and plays a central role in organ development by promoting proliferation, growth, migration, and differentiation (*Hogan and Schulte-Merker, 2017*; *Lavoie et al., 2020*). As such, Erk-signalling must be tightly regulated in both its spatial and temporal activation. To understand how dynamically Erk activity is regulated in developing vasculature, we generated the EC-EKC transgenic line and validated its use as a proxy readout of active Erk-signalling in vasculature. We found that it both provided a valid readout for physiological Erk-signalling and uncovered previously unappreciated Erk-signalling dynamics during vessel regeneration (*Figure 7*). In the context of tip-cell proliferation in angiogenesis, we revealed a very rapid post-cell division signalling asymmetry, confirming a previous work based on static imaging (*Costa et al., 2016*). In regenerative angiogenesis, we revealed a two-step mechanism for Erk-signalling activation post-vessel wounding that involves an immediate and ongoing signalling response that progressively limits Erk-signalling to vessels that are regenerating. Importantly, this study shows the utility of this new transgenic line to elucidate dynamic Erk-signalling events in vertebrate ECs and we suggest it will be a useful tool for diverse future studies of development and disease.

At the technical level, we used various quantification methods for measuring Erk activity in ECs and all generated valid results. The ratio of nuclear/cytoplasm EKC localisation gives the most accurate readout (*Regot et al., 2014*), but can only be used when a cell's cytoplasmic fluorescence can be accurately measured. This is especially challenging for ECs that overlap and have an unpredictable morphology in vascular tubes. *de la Cova et al., 2017* used a second-generation ERK KTR, which includes a nuclear localised H2B-mCherry expressed from the same

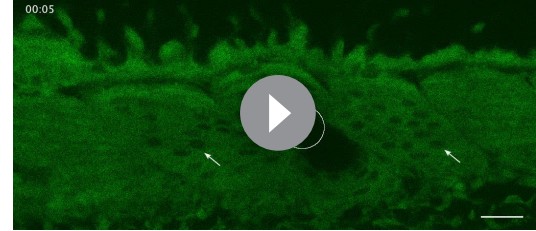

**Video 6.** Skin epithelial and muscle cells do not maintain high Erk activity for 3 hr following muscle wounding. Time-lapse video of the trunk in a 30 hours post-fertilisation (hpf) *Tg(ubb:Mmu.Elk1-KTR-mCherry)* embryo following muscle wounding. The white circle shows the wounded site. Skin epithelial and muscle cells surrounding the wounded site do not sustain Erk activity (examples of Erk-active cells, with nuclear excluded EKC expression indicated with white arrows). Z stacks were acquired nevery 21 min from 5 min post-ablation (mpa) until 3 hours post-ablation (hpa) using a Leica SP8 X WLL confocal microscope (n = 6 embryos). Scale bar: 20 µm.
https://elifesciences.org/articles/62196#video6

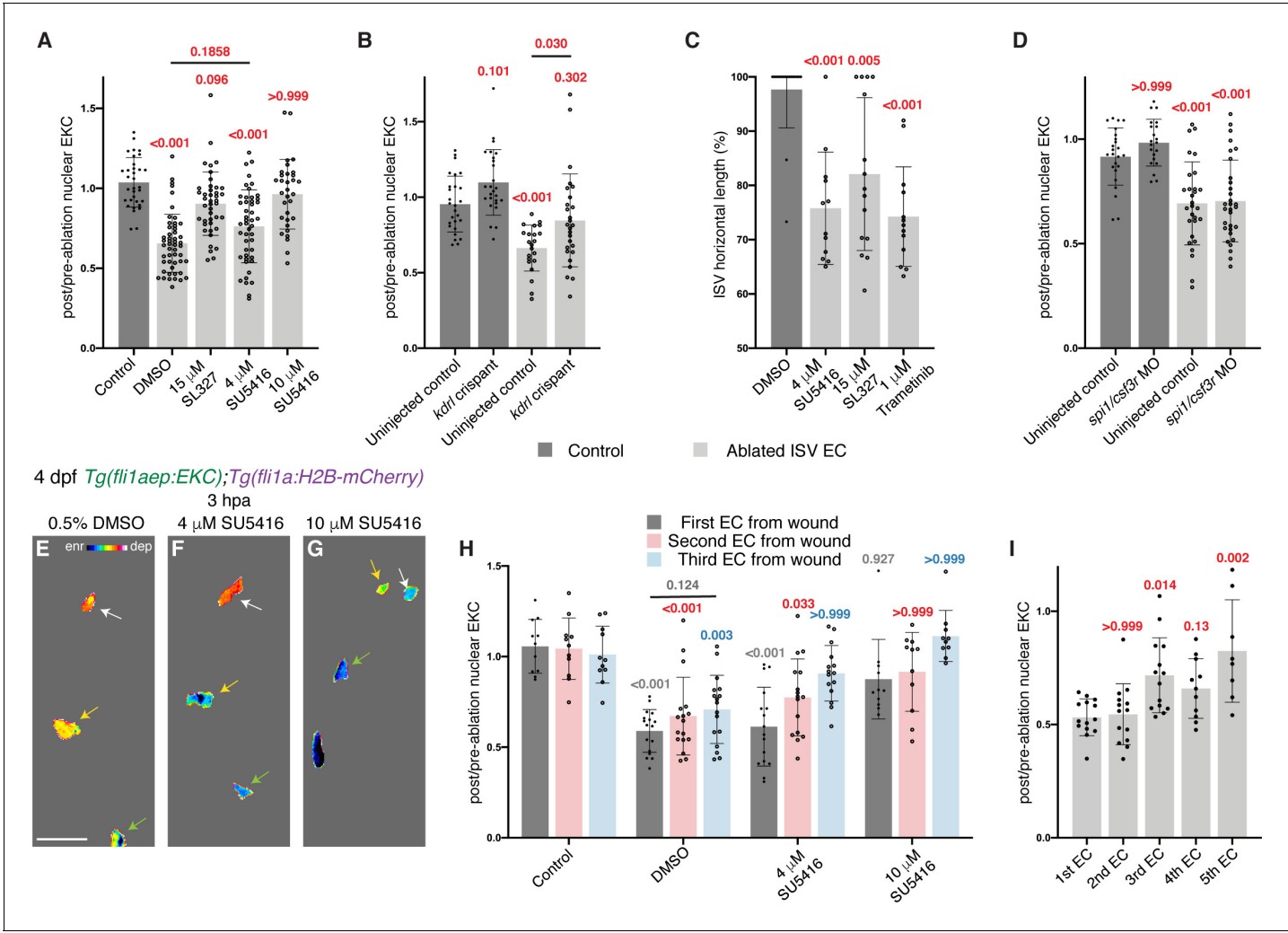

**Figure 5.** Erk activity in ablated vessels is maintained through the Vegfr pathway. (**A**) Ongoing Erk-signalling requires Vegfr and Mek activity. Quantification of post-/pre-ablation nuclear EKC intensity 3 hours post-ablation (hpa) in endothelial cells (ECs) of 0.5% dimethyl sulfoxide (DMSO)-treated non-ablated control ISVs (33 ECs, n = 11 larvae) and ablated ISVs of larvae treated with either 0.5% DMSO (51 ECs, n = 17 larvae), 15 µM SL327 (42 ECs, n = 14 larvae), 4 µM SU5416 (47 ECs, n = 16 larvae), or 10 µM SU5416 (32 ECs, n = 11 larvae). (**B**) Kdrl is required for full induction of Erk activity in ablated ISV ECs. Quantification of post-/pre-ablation nuclear EKC intensity 3 hpa in non-ablated control ISV ECs of uninjected control (27 ECs, n = 9 larvae) and *kdrl* crispants (26 ECs, n = 9 larvae), and ablated ISV ECs of uninjected control (22 ECs, n = 8 larvae) and *kdrl* crispants (27 ECs, n = 9 larvae). (**C**) Quantification of ISV horizontal length (as percentage of control) for ablated ISVs in 24 hpa, 5 days post-fertilisation (dpf), EC-EKC larvae treated with either 0.5% DMSO (n = 18 larvae), 4 µM SU5416 (n = 12 larvae), 15 µM SL327 (n = 15 larvae), or 1 µM Trametinib (n = 13 larvae). (**D**) Macrophages are not required for maintaining Erk activity in ablated ISV ECs. Quantification of post-/pre-ablation nuclear EKC intensity 3 hpa in non-ablated control ISV ECs of uninjected control (24 ECs, n = 8 larvae) and *spi1/csf3r* morphants (21 ECs, n = 7 larvae), and ablated ISV ECs of uninjected control (29 ECs, n = 10 larvae) and *spi1/csf3r* morphants (31 ECs, n = 11 larvae). (**E–G**) Lateral spinning disc confocal images of ablated ISV ECs in 4 dpf, 3 hpa, EC-EKC larvae treated with either 0.5% DMSO (**E**), 4 µM SU5416 (**F**), or 10 µM SU5416 (**G**). EC Erk activity was consistently higher and more Vegfr-dependent closer to the wound. Arrows indicate first (white), second (yellow), and third (green) ECs from the wounded site. Full images: *Figure 5—figure supplement 1D',H',J'*. (**H**) Quantification of post-/pre-ablation nuclear EKC intensity at 3 hpa in first (dark grey), second (red), and third (light blue) ECs from wound. Treatments were 0.5% DMSO-treated non-ablated control ISVs (11 first, second, and third ECs, n = 11 larvae), and ablated ISVs of larvae treated with either 0.5% DMSO (17 first, second, and third ECs, n = 17 larvae), 4 µM SU5416 (16 first and second ECs, and 15 third ECs, n = 16 larvae), or 10 µM SU5416 (11 first and second ECs, and 10 third ECs, n = 11 larvae). The same embryos were used in (**A**). (**I**) Quantification of post-/pre-ablation nuclear EKC intensity at 3 hpa in first (14 ECs, n = 14 larvae), second (14 ECs, n = 14 larvae), third (14 ECs, n = 14 larvae), forth (11 ECs, n = 11 larvae), and fifth (8 ECs, n = 8 larvae) ECs from the wounded site of ablated ISVs in 4 dpf EC-EKC larvae. Data for the first, second, and third ECs were taken from *Figure 4—figure supplement 1N*. ISV: intersegmental vessel; DA: dorsal aorta. Statistical test: Kruskal-Wallis test was conducted for graphs (**A, C, D, H, I**). Ordinary one-way ANOVA test was conducted for graph (**B**). Error bars represent standard deviation. 15 µm for image (**E**).

The online version of this article includes the following source data and figure supplement(s) for figure 5:

**Source data 1.** Post-/pre-ablation nuclear EKC measurements in control, ablated, and adjacent ISV ECs at 3 hpa.

*Figure 5 continued on next page*

promoter, allowing them to quantify Erk activity based on the Clover/mCherry ratio in *Caenorhabditis elegans*. We used a similar approach here with two independent transgenes driving EKC and H2B-mCherry and produced highly consistent results. It is worth noting that inter-embryo/larvae variations in H2B-mCherry intensity need to be considered, hence transgenic lines that express both ERK KTR and H2B-mCherry under a single promoter would be ideal. Finally, we also used the measurement of nuclear EKC normalised to the average EKC intensity of the DA to normalise for embryo to embryo variation. This approach also provided data consistent with the other two methods. Thus, overall, this EC-EKC model is highly robust with multiple methods to quantify and normalise sensor localisation. As KTR reporters are used more frequently in vivo in the future, the quantification methods used here may be applied to many scenarios analysing cellular Erk activity in cells with a complex 3D morphology.

Studies in zebrafish and *Xenopus* have demonstrated rapid Erk activation in epithelial cells upon local wounding, which subsides relatively quickly (within 1 hpa) as tissue repair progresses (*Li et al., 2013*; *Mayr et al., 2018*). Interestingly, our work shows a similar, very rapid, Erk activation in all vasculature in proximity to a wound. This suggests a common, initial, rapid Erk-signalling response immediately post-wounding in many different cell types and tissues – as if cells adjacent to a wound are rapidly primed to respond. However, in the vasculature, this signalling returned to pre-ablation levels by 1 hpa, while Erk activity was maintained for a longer timeframe only in the wounded vessels. This ongoing, later signalling was maintained through Vegfr activity, likely stimulated in part by Vegfa secreted from macrophages (*Gurevich et al., 2018*), and our data suggests other local sources of Vegfs (see *Figure 7*). Thus, Erk-signalling dynamics between wounded (ablated) and unwounded (adjacent) vessels differed significantly. We suggest this difference represents an initial priming of the wounded tissue (the rapid Erk response) that is replaced over time with sustained vascular Erk-signalling that is essential in the regenerative response.

Rapid $Ca^{2+}$ signalling post-wounding is observed in multiple systems in vitro and in vivo (reviewed in detail in *Ghilardi et al., 2020*). Using both quantitative live-imaging and pharmacological inhibition, we found that $Ca^{2+}$ signalling is required for Erk activation in ablated ISV ECs. Taking advantage of the high spatial and temporal resolution

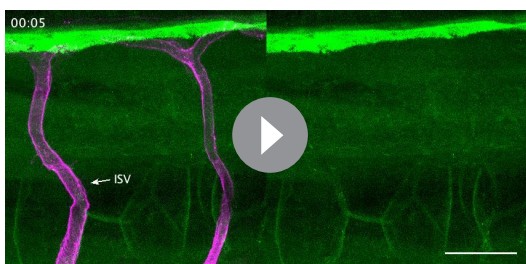

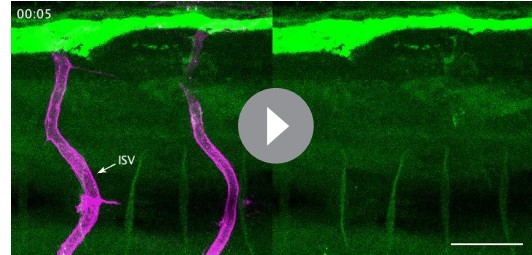

**Video 7.** ISVs in 4 dpf larvae do not have active $Ca^{2+}$ signalling. Time-lapse video of ISVs in a 4 days post-fertilisation (dpf) *Tg(actb2:GCaMP6f);Tg(kdrl:mCherry-CAAX)* larva. Functional vessels at 4 dpf have low or undetectable $Ca^{2+}$ signalling. Left panel shows both the *actb2:GCaMP6f* and the *kdrl:mCherry-CAAX* expression and the right panel shows the *actb2:GCaMP6f* expression. Z stacks were acquired every minute for 15 min using a Leica SP8 confocal microscope. ISV: intersegmental vessel. Scale bar: 50 μm.
https://elifesciences.org/articles/62196#video7

**Video 8.** ISVs rapidly activate $Ca^{2+}$ signalling following vessel wounding. Time-lapse video of both ablated and adjacent ISVs in a 4 days post-fertilisation (dpf) *Tg (actb2:GCaMP6f);Tg(kdrl:mCherry-CAAX)* larva following vessel wounding. Only the wounded ISV activates $Ca^{2+}$ signalling. Left panel shows both the *actb2:GCaMP6f* and the *kdrl:mCherry-CAAX* expression, and the right panel shows the *actb2:GCaMP6f* expression. Z stacks were acquired every minute from 5 min post-ablation (mpa) until 20 mpa using a Leica SP8 confocal microscope. ISV: intersegmental vessel. Scale bar: 50 μm.
https://elifesciences.org/articles/62196#video8

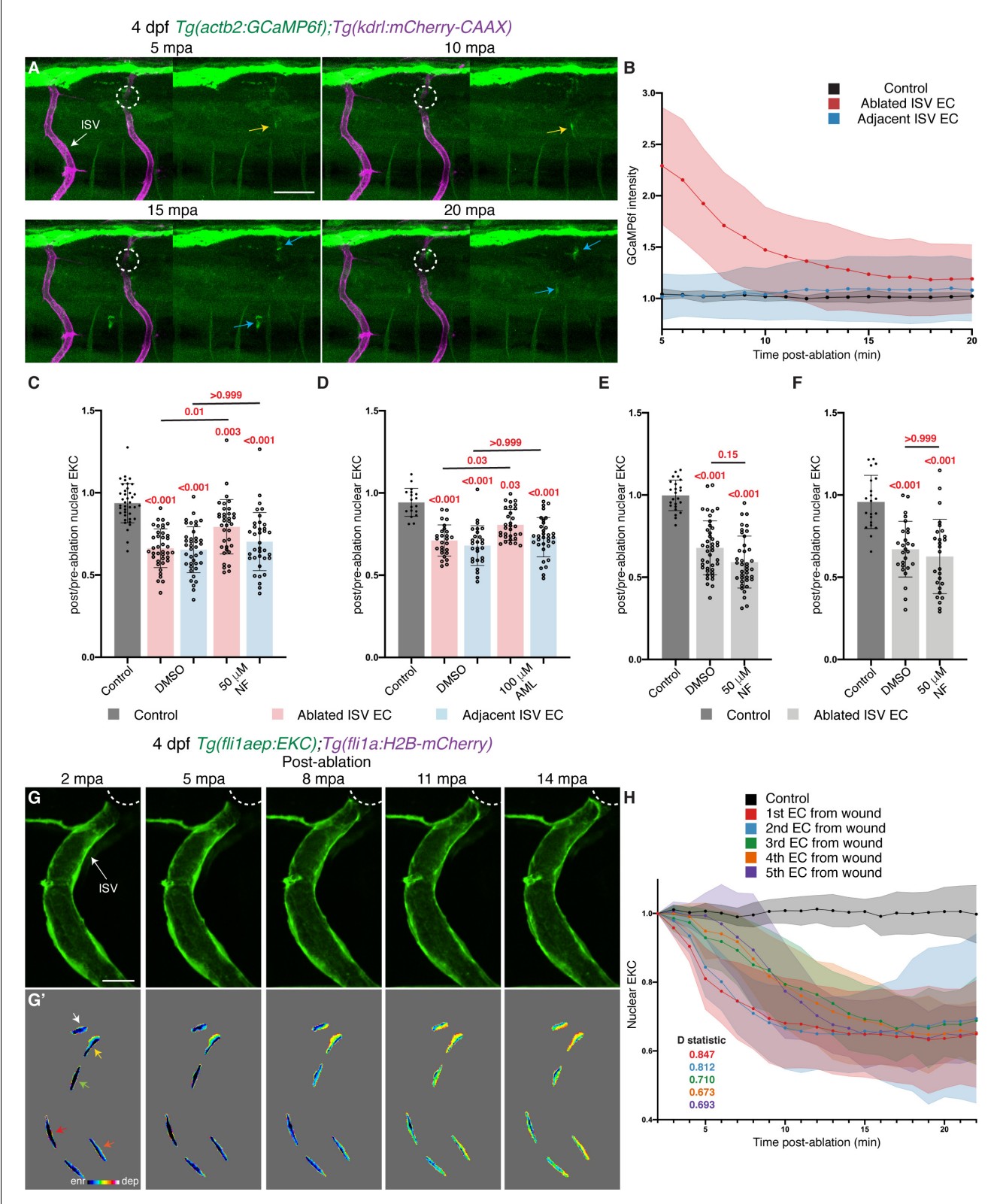

**Figure 6.** Ca$^{2+}$ signalling is required for rapid Erk activation in ablated vessels. (**A**) Still images from *Video 8* demonstrating a pulse of Ca$^{2+}$ signalling immediately adjacent to the wound (4 days post-fertilisation (dpf)). Left panels show *actb2:GCaMP6f* and *kdrl:mCherry-CAAX*, and right panels show *actb2:GCaMP6f*. Yellow arrows show ISV endothelial cells (ECs) with active Ca$^{2+}$ signalling. Blue arrows show Ca$^{2+}$ signalling in recruited immune cells. (**B**) Quantification of *actb2:GCaMP6f* intensity in unablated control ISVs (black, n = 4 larvae), ablated (red, n = 10 larvae) and adjacent (light blue, n = 10

*Figure 6 continued on next page*

*Figure 6 continued*

larvae) ISVs following wounding. Intensity was normalised to *actb2:GCaMP6f* intensity in unablated tissue in the same larvae. (C) Ca$^{2+}$ signalling is required for rapid activation of Erk-signalling in ablated ISV ECs. Quantification of post-/pre-ablation nuclear EKC intensity at 15 min post-ablation (mpa) in ECs of 1% dimethyl sulfoxide (DMSO)-treated non-ablated control ISVs (39 ECs, n = 13 larvae), and ISVs of larvae treated with either 1% DMSO (39 ablated/adjacent ISV ECs, n = 13 larvae) or 50 µM Nifedipine (36 ablated/adjacent ISV ECs, n = 12 larvae). (D) Quantification of post-/pre-ablation nuclear EKC intensity at 15 mpa in ECs of 1% DMSO-treated non-ablated control ISVs (18 ECs, n = 6 larvae), and ISVs of larvae treated with either 1% DMSO (27 ablated/adjacent ISV ECs, n = 9 larvae) or 100 µM Amplopidine (31 ablated ISV ECs and 33 adjacent ISV ECs, n = 11 larvae). (E) Ca$^{2+}$ signalling is not required for sustaining Erk activity in ablated ISV ECs. Quantification of post-/pre-ablation nuclear EKC intensity at 3 hours post-ablation (hpa) in ECs of 1% DMSO-treated non-ablated control ISVs (24 ECs, n = 8 larvae), and ablated ISVs of larvae treated with either 1% DMSO (42 ECs, n = 14 larvae) or 50 µM Nifedipine (39 ECs, n = 13 larvae) for 30 min before 3 hpa (*Figure 6—figure supplement 2A*). (F) Quantification of post-/pre-ablation nuclear EKC intensity at 3 hpa in ECs of 1% DMSO-treated non-ablated control ISVs (21 ECs, n = 7 larvae), and ablated ISVs of larvae treated with either 1% DMSO (27 ECs, n = 9 larvae) or 50 µM Nifedipine (27 ECs, n = 9 larvae) for 30 min after vessel wounding (*Figure 6—figure supplement 2H*). (G, G′) Still images from *Video 3* showing ablated ISV ECs of a 4 dpf EC-EKC larva after vessel wounding. Activation of Erk progresses from the wound to the vessel base. Image (G) shows *fli1aep:EKC* expression, and (G′) shows nuclear *fli1aep:EKC* intensity. Arrows indicate first (white), second (yellow), third (green), forth (red), and fifth (orange) ECs from the wounded site. (H) Quantification of nuclear EKC intensity (normalised to nuclear EKC intensity at 2 mpa) in ECs of ISVs in non-ablated control larvae (black, 24 ECs, n = 8 larvae), and the first (red, 9 ECs, n = 9 larvae), second (blue, 9 ECs, n = 9 larvae), third (green, 9 ECs, n = 9 larvae), fourth (orange, 8 ECs, n = 8 larvae), and fifth (purple, 5 ECs, n = 5 larvae) ablated ISV ECs from the wounded site following vessel wounding. ISV: intersegmental vessel. Statistical test: Kruskal-Wallis test was conducted for graphs (C-F). Two-sample Kolmogorov-Smirnov test was conducted for graph (H). Error bars represent standard deviation. Scale bars: 50 µm for image (A), 15 µm for image (G).

The online version of this article includes the following source data and figure supplement(s) for figure 6:

**Source data 1.** GCaMP6f intensity measurements and post-/pre-ablation nuclear EKC measurements in control, ablated, and adjacent ISV ECs.
**Figure supplement 1.** Ca$^{2+}$ signalling is required for rapid Erk activation in ablated ISV ECs.
**Figure supplement 2.** Ca$^{2+}$ signalling is not required for sustained Erk activation in ablated ISV ECs.

---

in our model, we found that Ca$^{2+}$-dependent Erk-signalling is activated progressively from cells closest to the wound to cells farther away. This may be consistent with a wave of Ca$^{2+}$ signalling through the wounded vessel. Activation of Erk-signalling at 2 mpa in wounded epithelial cells in *Xenopus* promotes actomyosin contraction and wound closure (*Li et al., 2013*). Therefore, rapid Ca$^{2+}$ signalling-mediated Erk activation in the wounded vessel may ensure efficient wound closure in ablated ISVs. At a molecular mechanistic level, it seems likely that EC Ca$^{2+}$ signalling is influenced by the activity of either transient receptor potential (TRP) channels (*Smani et al., 2018*) or P2X receptors (P2X4 or P2X7) (*Surprenant and North, 2009*), which are active in ECs and can influence angiogenesis, cytoskeletal remodelling, and vascular permeability. We found no evidence that Ca$^{2+}$ signalling influenced the broader, rapid Erk-signalling response in unwounded but adjacent vasculature. One interesting candidate to contribute to this broader mechanism is altered tissue tension associated with the tissue ablation, which had been shown in some contexts to modulate ERK-signalling (*Rosenfeldt and Grinnell, 2000*; *Hirata et al., 2015*). Perhaps consistent with this idea, we did not identify a mechanism required for rapid Erk activation in adjacent ISV ECs and vessel wounding was not required – tissue wounding in between ISVs alone activated Erk-signalling in surrounding ECs. Further work is needed to fully appreciate the role of mechanical contributions in this response. Nevertheless, rapid Erk activation in ECs upon wounding seems likely to potentiate these ECs to more rapidly respond to external growth factors such as Vegfa upon the later activation of the inflammatory response and initiation of sustained regenerative angiogenesis.

Taking advantage of spatial information in the imaging data, we showed that ECs in wounded ISVs that are actively regenerating at 3 hpa display a graded signalling response along the vessel at the level of Vegfr/Erk activity. This is likely due to a local source (or sources) of Vegfa and may explain why unwounded ISV ECs, which are farther away from the Vegfa source, do not sustain high Erk activity at 3 hpa. In bigger wounds, excessive angiogenesis has been previously reported to occur from adjacent ISVs, and macrophage-dependent vascular regression is then required to ensure vessel patterns return to their original state (*Gurevich et al., 2018*). Therefore, we hypothesise that maintaining Erk activity only in ECs of vessels that need to regenerate in this laser ablation model ensures EC proliferation, and migration only occurs in regenerating vessels and prevents excessive angiogenesis. Further studies could investigate Erk-signalling dynamics of ECs in bigger wounds, which more closely resemble traumatic injuries in humans, and could further assess Erk-signalling dynamics in excessive angiogenesis and regression.

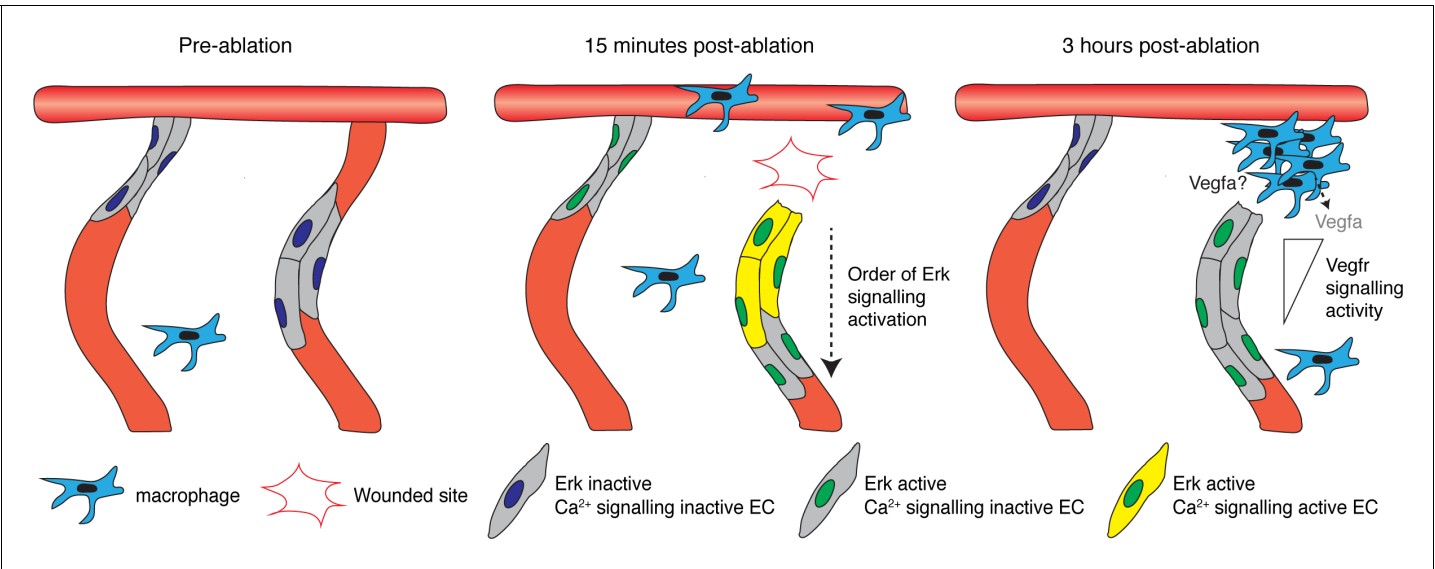

**Figure 7.** A two-step mechanism for activating and maintaining Erk activity in regenerating vessels. Schematic representation of the two-step mechanism employed by endothelial cells (ECs) to activate Erk-signalling following vessel wounding. Pre-ablation (left), the majority of ECs are Erk-signalling inactive. Following vessel wounding (middle), both ablated and adjacent intersegmental vessel (ISV) ECs rapidly activate Erk-signalling. $Ca^{2+}$ signalling is also rapidly activated following vessel wounding but only in ablated ISV ECs, particularly in ECs close to the wounded site. $Ca^{2+}$ signalling activity contributes to the activation of Erk-signalling in ablated ISV ECs in a sequential manner, starting from ECs close to the wounded site. Erk-signalling in adjacent ISV ECs has returned to pre-wound levels by 3 hours post-ablation (hpa) (right). Erk activity in ablated vessel ISV ECs is sustained through Vegfr-signalling. ECs closer to the wounded site are less sensitive to Vegfr-signalling inhibition, with higher signalling compared to ECs farther away. Recruited macrophages are essential for vessel regeneration but not the sole source of Vegfs at 3 hpa.

Blood vessels constantly remodel to accommodate for the needs of the human body during development and disease (*Carmeliet and Jain, 2011*; *Chung and Ferrara, 2011*; *Potente et al., 2011*). It is therefore not surprising that Erk-signalling, which is a key modulator of angiogenesis, is highly dynamic in ECs. As a novel tool that allows real-time analysis of Erk activity, EC-EKC biosensors will be useful for elucidating Erk-signalling events in vasculature in an array of settings and different vertebrate models. Importantly, in zebrafish, the EC-EKC transgenic line can be coupled with both established and novel mutants with vascular phenotypes to investigate how real-time EC Erk-signalling dynamics is affected in the absence of key vascular genes. Further, dynamic Erk-signalling events in ECs in zebrafish disease models associated with increased angiogenesis such as in cancer (*Nicoli et al., 2007*) and tuberculosis (*Oehlers et al., 2015*) can be analysed using this EC-EKC model. This could highlight novel pathological Erk-signalling events in ECs, which could be normalised using drugs shown to modulate Erk-signalling (*Goglia et al., 2020*). Of note, KTR constructs for other kinases such as AKT, JNK, and p38 are also now available (*Regot et al., 2014*; *Maryu et al., 2016*). Other types of fluorescence-based kinase activity reporters, such as separation of phases-based activity reporter of kinases (SPARK), could also be applied (*Zhang et al., 2018*). Future studies will inevitably combine multiple signalling biosensors to elucidate real-time interactions between signalling pathways as they decipher incoming signals and drive development and disease.

## Materials and methods

### Key resources table

| Reagent type (species) or resource | Designation | Source or reference | Identifiers | Additional information |
|---|---|---|---|---|
| Genetic reagent (*Danio rerio*) | *Tg(fli1a:H2B-mCherry)$^{uq37bh}$* | *Baek et al., 2019* | RRID:ZFIN_ZDB-ALT-191011-5 | Ben M Hogan (Organogenesis and Cancer Program, Peter MacCallum Cancer Centre, Australia) |

*Continued on next page*

*Continued*

| Reagent type (species) or resource | Designation | Source or reference | Identifiers | Additional information |
|---|---|---|---|---|
| Genetic reagent (*D. rerio*) | *Tg(fli1a:EGFP)*[y1] | *Lawson and Weinstein, 2002* | RRID:ZFIN_ZDB-ALT-011017-8 | Brant M Weinstein (National Institute of Child Health and Human Development, Bethesda, USA) |
| Genetic reagent (*D. rerio*) | *Tg(fli1aep:ERK-KTR-Clover)*[uq39bh] | This study | | Ben M Hogan (Organogenesis and Cancer Program, Peter MacCallum Cancer Centre, Australia) |
| Genetic reagent (*D. rerio*) | *Tg(ubb:Mmu.Elk1-KTR-mClover)*[vi1] | *Mayr et al., 2018* | ZFIN ID: ZDB-ALT-190211–6 | Martin Distel (Children's Cancer Research Institute, Austria) |
| Genetic reagent (*D. rerio*) | *Tg(actb2:GCaMP6f)*[zf3076] | *Herzog et al., 2019* | ZFIN ID: ZDB-ALT-200610–2 | Leah Herrgen (Centre for Discovery Brain Sciences, University of Edinburgh, Germany) |
| Genetic reagent (*D. rerio*) | *Tg(kdrl:mCherry-CAAX)*[y171] | *Fujita et al., 2011* | RRID:ZFIN_ZDB-ALT-110429-3 | Brent M Weinstein (National Institute of Child Health and Human Development, Bethesda, USA) |
| Genetic reagent (*D. rerio*) | *Tg(mpeg1:mCherry)*[gl23] | *Ellett et al., 2011* | RRID:ZFIN_ZDB-ALT-120117-2 | Graham Lieschke (Australian Regenerative Medicine Institute, Monash University, Australia) |
| Genetic reagent (*D. rerio*) | *Tg(kdrl:EGFP)*[s843] | *Beis et al., 2005* | RRID:ZFIN_ZDB-ALT-050916-14 | Didier Stainier (Max Planck Institute for Heart and Lung Research, Germany) |
| Sequence-based reagent | MO1-spi1b | *Rhodes et al., 2005* | ZFIN ID: ZDB-MRPHLNO-050224–1 | Genetools, LLC, OR, USA |
| Sequence-based reagent | MO3-csf3r | *Ellett et al., 2011* | ZFIN ID: ZDB-MRPHLNO-111213–1 | Genetools, LLC, OR, USA |
| Software, algorithm | FIJI | ImageJ (http://imagej.nih.gov/ij/) | RRID:SCR_002285 | Image processing and analysis, Version Fiji version 1 |
| Software, algorithm | Imaris x64 | Bitplane, Belfast, UK | RRID:SCR_007370 | Image processing and analysis, Version 9.5.1 |
| Software, algorithm | GraphPad Prism | GraphPad Prism (http://graphpad.com) | RRID:SCR_002798 | Statistics, Prism8: Version 8.3.0 |
| Software, algorithm | R/R Studio | R project (r-project.org) | RRID:SCR_001905 | Statistics, R version 4.0.2 |
| Chemical compound, drug | SL327 (MEK signalling inhibitor) | Merck, Darmstadt, Germany | S4069 | Diluted in DMSO |
| Chemical compound, drug | Trametinib (MEK signalling inhibitor) | Selleck chemicals, TX, USA | S2673 | Diluted in DMSO |
| Chemical compound, drug | SU5416 | Merck, Darmstadt, Germany | S8442 | Diluted in DMSO |
| Chemical compound, drug | AV951 | Adooq Bioscience, CA, USA | 475108-18-0 | Diluted in DMSO |
| Chemical compound, drug | Nifedipine | Bio-Techne, MN, USA | 1075 | Diluted in DMSO |
| Chemical compound, drug | Amlodipine | Merck, Darmstadt, Germany | A5605 | Diluted in DMSO |

## Zebrafish

All zebrafish works were conducted in accordance with the guidelines of the animal ethics committees at the University of Queensland (AE54297), University of Melbourne, Peter MacCallum Cancer Centre (E634 and E643), University of Bristol (3003318), and the Children's Cancer Research Institute (GZ:565304/2014/6 and GZ:534619/2014/4). The transgenic zebrafish lines used were published previously as following: *Tg(fli1a:H2B-mCherry)*[uq37bh] (*Baek et al., 2019*), *Tg(fli1a:EGFP)*[y1] (*Lawson and Weinstein, 2002*), *Tg(ubb:Mmu.Elk1-KTR-mClover)*[vi1] (*Mayr et al., 2018*), *Tg(actb2:GCaMP6f)*[zf3076]

(*Herzog et al., 2019*), *Tg(kdrl:mCherry-CAAX)$^{y171}$* (*Fujita et al., 2011*), *Tg(mpeg1:mCherry)$^{gl23}$* (*Ellett et al., 2011*), and *Tg(kdrl:EGFP)$^{s843}$* (*Beis et al., 2005*). The *Tg(fli1aep:ERK-KTR-Clover)$^{uq39bh}$* transgenic line (referred to as *Tg(fli1aep:EKC)*/EC-EKC in this study) was generated for this study using Gateway cloning and transgenesis. The pENTR-ERKKTRClover plasmid (#59138) was purchased from Addgene.

## Live-imaging and laser-inflicted vessel/tissue wounding

Embryos/larvae at indicated stages were immobilised with tricaine (0.08 mg/ml) and mounted laterally in either 1.2% ultra-low gelling agarose (specifically for *Video 6*), 0.25% low melting agarose (specifically for *Videos 7* and *8*, and *Figure 6A*), or 0.5% low melting agarose (Merck, Darmstadt, Germany; A9414-100G) as previously described (*Okuda et al., 2018*). Images were taken at indicated timepoints/timeframes using either a Zeiss LSM 710 confocal microscope using either a Zeiss Plan Apochromat X10 objective (dry, N.A. 0.45, specifically for *Figure 1B–E*) or a Zeiss Plan Apochromat X20 objective (dry, N.A. 0.8, specifically for *Figure 3A,B*), a Zeiss Elyra 780 confocal microscope using either a Zeiss Apochromat x10 objective (dry, N.A. 0.45, specifically for *Figure 5— figure supplement 1K,L*) or a Zeiss Plan Apochromat x40 objective (water, N.A. 1.1, specifically for *Figure 3—figure supplement 1A–B′*, *Figure 3—figure supplement 2H,I*, *Figure 5—figure supplement 1M–T′*, *Figure 5—figure supplement 2A–H′*, and *Figure 6—figure supplement 2I–N′*), a Leica SP8 X WLL confocal microscope using a Leica HC PL APO CS2 x40 objective (water, N.A. 1.1, specifically for *Video 6*), a Leica TCS SP8 multiphoton microscope using a Leica HC Fluotar x25 objective (water, N.A. 0.95, specifically for *Videos 7* and *8*, and *Figure 6A*), an Olympus Yokogawa CSU-W1 Spinning Disc Confocal microscope using a UPLSAPO x40 objective (silicon, N.A. 1.25, specifically for *Figure 6—figure supplement 1K–T′*), or an Andor Dragonfly Spinning Disc Confocal microscope using a Nikon Apo λ LWD x40 objective (water, N.A. 1.15).

Muscle wounding in 30 hpf *Tg(ubb:Mmu.Elk1-KTR-mClover)* embryos was conducted as previously described (specifically for *Video 6*; *Mayr et al., 2018*). Briefly, a laser-inflicted wound was introduced on mounted embryos using the Leica SP8 X FRAP module with the UV laser line of 405 nm at 85% laser power. Vessel wounding in 4 dpf *Tg(actb2:GCaMP6f);Tg(kdrl:mCherry-CAAX)* larvae was conducted as previously described (specifically for *Videos 7* and *8*, and *Figure 6A*; *Gurevich et al., 2018*). Briefly, a laser-inflicted wound was introduced on mounted larvae using a Micropoint laser (Spectra-Physics, CA, USA) connected to a Zeiss Axioplan II microscope with a laser pulse at a wavelength of 435 nm. All other tissue/vessel woundings in either 3 dpf (specifically for *Figure 3—figure supplement 2B,C,H,I,P,R,T,V* and *Figure 5—figure supplement 2F,H*) or 4 dpf *Tg(fli1aep:EKC);Tg(fli1a:H2B-mCherry)* or *Tg(kdrl:EGFP);Tg(mpeg1:mCherry)* larvae were conducted using either a Zeiss LSM 710 confocal microscope or an Olympus FVMPE-RS multiphoton microscope. Briefly, a laser-inflicted wound was introduced on mounted larvae using a two-photon laser at 790 nm (Zeiss LSM 710 confocal microscope) or 900 nm (Olympus FVMPE-RS multiphoton microscope) at 80% laser power (Mai Tai, Spectra-Physics, CA, USA). The area of laser ablation for vessel-wounding experiments was made consistent for all experiments (height: 40 μm, width: 15 μm). All vessel woundings were conducted on the ISV dorsal to the cloaca.

For *Video 1*, time-lapse images of ISVs in 24–25 *Tg(fli1aep:EKC);Tg(fli1a:H2B-mCherry)* embryos were acquired every 14–17 s for 40 min using an Andor Dragonfly Spinning Disc Confocal microscope. Difference in time intervals was due to difference in z section number in different embryos. Pre-division ISV tip ECs with cytoplasmic H2B-mCherry localisation were selected for imaging. For *Videos 3–5*, time-lapse images of ISVs in 4 dpf *Tg(fli1aep:EKC);Tg(fli1a:H2B-mCherry)* larvae were taken every minute for 20 min using an Andor Dragonfly Spinning Disc Confocal microscope, wounded as described above using a Zeiss LSM 710 confocal microscope, transferred to an Andor Dragonfly Spinning Disc Confocal microscope (allowing for 2 min to transfer the larvae and initiate imaging), and re-imaged every minute for another 20 min. As a control (*Video 2*), time-lapse images of ISVs in 4 dpf *Tg(fli1aep:EKC);Tg(fli1a:H2B-mCherry)* larvae were taken every minute for 41 min. For *Video 6*, time-lapse images of the trunk in a 30 hpf *Tg(ubb:Mmu.Elk1-KTR-mCherry)* embryo were acquired every 21 min from 5 mpa until 3 hpa using a Leica SP8 X WLL confocal microscope. For *Video 8*, time-lapse images of ISVs in 4 dpf *Tg(actb2:GCaMP6f);Tg(kdrl:mCherry-CAAX)* larvae were acquired every minute from 5 mpa until 20 mpa using a Leica SP8 confocal microscope. As a control (*Video 7*), time-lapse images of ISVs in 4 dpf *Tg(actb2:GCaMP6f);Tg(kdrl:mCherry-CAAX)* larvae were acquired every minute for 15 min using a Leica SP8 confocal microscope.

## Morpholino injections

The *spi1b* and *csf3r* morpholinos used in this study have been validated and described previously (*Rhodes et al., 2005*; *Ellett et al., 2011*; *Pase et al., 2012*). A cocktail of *spi1b* (5 ng) and *csf3r* (2.5 ng) morpholinos was injected into one- to four-cell-stage EC-*Tg(fli1aep:EKC);Tg(fli1a:H2B-mCherry)* or *Tg(mpeg1:mCherry)* embryos as previously described (*Pase et al., 2012*). ISVs of 3 dpf morphants/uninjected controls were imaged before vessel wounding, wounded as described above, and reimaged either at 15 mpa or at 3 hpa. To measure vessel regeneration, ISVs of 3 dpf morphants/uninjected controls were wounded as described above and imaged at 24 hpa. Non-ablated 3 dpf *Tg(fli1aep:EKC);Tg(fli1a:H2B-mCherry)* morphants/uninjected controls were imaged and re-imaged either 15 min or 3 hr later. Macrophage numbers (*mpeg1:mCherry*-positive) in 3 dpf embryos (*Figure 3—figure supplement 2E,F*) or 4 dpf larvae (*Figure 3—figure supplement 2A–C*) were manually quantified using the cell counter tool in FIJI.

## Drug treatments

For investigating Erk activity in ISV tip ECs in 28 hpf embryos following drug treatment, 27 hpf *Tg(fli1aep:EKC);Tg(fli1a:H2B-mCherry)* embryos were treated for an hour with either 0.5% DMSO (vehicle control), 15 µM SL327, 4 µM SU5416, or 500 nM AV951 diluted in E3 medium with 0.003% 1-phenyl-2-thiourea (PTU) and imaged as described above at 28 hpf. Up to five ISV tip ECs were quantified per embryo.

For investigating the role of prolonged EC Erk activity in vessel regeneration, ISVs of 4 dpf *Tg(fli1aep:EKC);Tg(fli1a:H2B-mCherry)* larvae were wounded as described above and were treated with either 0.5% DMSO (vehicle control), 4 µM SU5416, 15 µM SL327, or 1 µM Trametinib for 24 hr and imaged as described above at 5 dpf (24 hpa). For measuring Erk activity in ECs pre- and post-ablation in 4 dpf larvae following drug treatment, 4 dpf *Tg(fli1aep:EKC);Tg(fli1a:H2B-mCherry)* larvae were first treated for an hour with either 0.5% DMSO, 15 µM SL327, 4 or 10 µM SU5416, or 500 nM AV951. ISVs of these larvae were imaged and then wounded as described above in the presence of respective drugs at indicated concentrations in the mounting media. The same larvae were reimaged at 15 mpa. Alternatively, larvae were removed from mounting media following vessel wounding and incubated in respective drugs at indicated concentrations in E3 media, before being remounted and imaged at 3 hpa.

For Nifedipine and Amlopidine treatments, 4 dpf *Tg(fli1aep:EKC);Tg(fli1a:H2B-mCherry)* larvae were first treated for 30 min with either 1% DMSO, 50 µM Nifedipine, or 100 µM Amlodipine. This was because treatment for 1 hr with either 50 µM Nifedipine or 100 µM Amlodipine resulted in mortalities due to reduced cardiac function. The ISVs of these larvae were imaged and wounded as described above and reimaged 15 mpa. Alternatively, 4 dpf *Tg(fli1aep:EKC);Tg(fli1a:H2B-mCherry)* larvae were imaged before vessel wounding, and removed from mounting media following vessel wounding and incubated in 1% DMSO. 30 min before 3 hpa, larvae were treated with 50 µM Nifedipine or continued its treatment with 1% DMSO, before being remounted in the presence of respective drugs at indicated concentrations and reimaged 3 hpa. To treat the larvae for 30 min with 50 µM Nifedipine following vessel wounding, 4 dpf *Tg(fli1aep:EKC);Tg(fli1a:H2B-mCherry)* larvae were mounted with either 1% DMSO or 50 µM Nifedipine, imaged before vessel wounding, and removed from mounting 30 min following vessel wounding. These larvae were incubated in 1% DMSO and reimaged 3 hpa. Non-ablated 4 dpf *Tg(fli1aep:EKC);Tg(fli1a:H2B-mCherry)* larvae controls were imaged, then reimaged either 15 min or 3 hr later.

*kdrl* guide RNA (gRNA) sequences were designed previously (*Wu et al., 2018*) and synthesised with the following oligonucleotide primers: Kdrl gRNA oligonucleotide 1: TAATACGACTCACTA TAGGCTTTCTGGTTCGATGGCAGTTTTAGAGCTAGAAATAGC; Kdrl gRNA oligonucleotide 2: TAA TACGACTCACTATAGGCTGTAGAGACCCCTCTCCGTTTTAGAGCTAGAAATAGC; Kdrl gRNA oligonucleotide 3: TAATACGACTCACTATAGGCACTCATAGCCGAGTGTAGTTTTAGAGCTAGAAA TAGC; Kdrl gRNA oligonucleotide 4: TAATACGACTCACTATAGGGTCACACTGCTCATCGAGG TTTTAGAGCTAGAAATAGC. Guide RNAs were synthesised as described previously (*Gagnon et al., 2014*) with modifications. Briefly, *kdrl* gRNA oligonucleotides were annealed to a constant oligonucleotide, ssDNA overhangs were filled in with T4 DNA polymerase (New England Biolabs, Victoria, Australia), and gRNA templates were purified using the DNA Clean and Concentrator Kit (Zymo Research, D4014, CA, USA). *kdrl* four-guide RNA cocktail was transcribed with Ambion Megascript

T7 promoter kit and cleaned using the RNA clean and concentrator Kit (Zymo Research, R1014, CA, USA). One-cell-stage *Tg(fli1aep:EKC);Tg(fli1a:H2B-mCherry)* embryos were injected with a cocktail of Cas9 protein (Integrated DNA Technologies, 1081059, IA, USA) and the guide RNAs. Only *kdrl* crispants with clear vascular phenotypes (*Figure 5—figure supplement 1L*) were used for all experiments. ISVs of 4 dpf crispants/uninjected controls were imaged before vessel wounding, wounded as described above, and reimaged at 3 hpa. Non-ablated 4 dpf *Tg(fli1aep:EKC);Tg(fli1a:H2B-mCherry)* crispants/uninjected controls were imaged, and re-imaged 3 hr later. As vessel wounding often resulted in no ECs in ISVs, ECs of connecting horizontal myoseptum vessels were used for ablation and quantification (*Figure 5—figure supplement 1T*).

## Image processing and analysis

Images were processed with image processing software FIJI version 1 (*Schindelin et al., 2012*) and Imaris x64 (Bitplane, Version 9.5.1). Erk activity in ECs was measured by comparing either nuclear/cytoplasm EKC intensity, nuclear EKC/H2B-mCherry intensity, or nuclear EKC intensity. In figures, EC-EKC intensity in nuclei is represented after masking nuclear expression using H2B-mCherry and presenting EC-EKC intensity in 16 colour LUT (Fiji). The nuclear/cytoplasm EKC intensity was quantified as described before (*Kudo et al., 2018*) with modifications, using a semi-autonomous custom written script in the ImageJ macro language. Briefly, z stack images were first processed into a maximum intensity z-projection. H2B-mCherry-positive EC nuclei underwent thresholding and were selected as individual regions of interest (ROIs). The EKC channel was converted to a 32-bit image with background (non-cell associated) pixels converted to NaN. The average pixel intensity of EKC in the nuclei ROIs was measured (nuclear EKC intensity). Nuclei ROIs were then expanded and converted to a banded selection of the adjacent cytoplasmic area and the average pixel intensity of EKC within the expanded ROIs was measured (cytoplasm EKC intensity). The custom written ImageJ macro is available at https://github.com/NickCondon/Nuclei-Cyto_MeasuringScipt (*Okuda, 2021*; copy archived at swh:1:rev:8c4e8e4f02d21c6545f75b864aec63f823abcfe7).

The average pixel intensity of either nuclear EKC or H2B-mCherry of ECs in 3D was quantified using Imaris software. The entire EC nucleus was masked using the H2B-mCherry signal. *Figure 2J and K* represents averages of data within each minute. For embryos/larvae exposed to a long-term time-lapse (for example, *Videos 2–5*) or ablated with high-powered multiphoton laser for ablation studies, difference in photostability between fluorophores could significantly alter the ratio of nuclear EKC/H2B-mCherry intensity (*Lam et al., 2012*). Therefore, we either compared the ratio of nuclear EKC intensities between ECs within the same fish (for example, *Video 1*) or normalised EC nuclear EKC intensity with the average EKC intensity of another EKC-expressing structure (for example, *Videos 2–5*). For larvae that underwent laser-inflicted wounding, nuclear EKC intensity pre- and post-ablation was normalised with the average pixel intensity of EKC of the entire DA within two-somite length. The ROI that covers the same DA region in pre- and post-wounded larvae was manually selected on a maximum intensity z-projection of the EKC channel, and average pixel intensity was calculated using FIJI. Datasets were presented either as the ratio of post-/pre-ablation normalised nuclear EKC intensity or as normalised nuclear EKC intensity further normalised to normalised nuclear EKC intensity in 2 mpa ECs (specifically for *Figure 6H*). Three closest ECs from the wounded site in both ablated and adjacent ISVs were quantified, except for *Figures 5I* and *6H*, where five closest ECs from the wounded site in ablated ISVs were analysed. For *Videos 2–5*, reduction in EKC intensity due to photobleaching was minimised using the bleach correction tool (correction method: histogram matching) in FIJI; however, quantifications were all done using raw data.

GCaMP6f average pixel intensity on ISVs and unablated tissue in 4 dpf *Tg(actb2:GCaMP6f);Tg(kdrl:mCherry-CAAX)* larvae was measured using FIJI. Maximum intensity z-projection images of both GCaMP6f and mCherry-CAAX channels were first corrected for any drift in x/y dimensions. An ROI was drawn around the mCherry-CAAX-positive ISV segment nearest to the site of injury (an area consistently between 100 and 150 $\mu m^2$) and the average pixel intensity of GCaMP6f within the ROI at each timepoint was measured using FIJI. Similar measurements were acquired for adjacent ISVs, ISVs in unablated control larvae, and uninjured tissue, maintaining consistent size of ROI within each biological replicate. ISV GCaMP6f average pixel intensity was normalised to the average pixel intensity in uninjured tissue GCaMP6f within the same larvae.

Percentage ISV height was measured by dividing the actual horizontal height of the ISV with the prospective horizontal height of the ISV (the horizontal height from the base ISV/DA intersection to the prospective ISV/DLAV intersection). Ellipticity (elliptic) of ISV tip ECs was quantified using Imaris software. Original raw data with relevant acquisition metadata can be provided upon request.

## Statistics

Graphic representations of data and statistical analysis were performed using either Prism 8 Version 8.3.0 or R software. Mann-Whitney test was conducted when comparing two datasets and Kruskal-Wallis test was conducted when comparing multiple datasets using Prism 8 (except for *Figure 5B*, for which an ordinary one-way ANOVA test was conducted, following confirmation of normality of all datasets using Anderson-Darling, D'Agostino and Pearson, Shapiro-Wilk, and Kolmogorov-Smirnov tests). Natural permutation test (*Figure 3H* and *Figure 4C*) or two-sample Kolmogorov-Smirnov test (*Figure 6H*) was used to test for differences between the population mean curve for datasets using R statistical software. For *Figure 6H*, we applied the non-parametric two-sample Kolmogorov-Smirnov test to evaluate whether the distribution of Erk activity for each position differed from that of the control. Null hypothesis was rejected where the D-statistic (maximum difference between two empirical cumulative distribution function (ECDF)) exceeded the critical threshold (critical D) for each comparison and p-value<0.001. D-statistic indicates magnitude of change for each curve compared with control. Critical D varied for each position as follows: control vs first EC from wound, 0.166; control vs second EC from wound, 0.166; control vs third EC from wound, 0.166; control vs fourth EC from wound, 0.173; control vs fifth EC from wound, 0.209. A p-value below 0.05 was considered statistically significant for all data. Error bars in all graphs represent standard deviation.

## Acknowledgements

This work was supported by NHMRC grants 1164734 and 1165117. BMH was supported by an NHMRC fellowship 1155221. We thank Dr. Enid Lam for technical assistance. Imaging was performed in the Australian Cancer Research Foundation's Cancer Ultrastructure and Function Facility at IMB, Centre for Advanced Histology and Microscopy at Peter MacCallum Cancer Centre, Wolfson Bioimaging Facility at University of Bristol, and the Zebrafish platform Austria for preclinical drug screening at the Children's Cancer Research Institute supported by the Austrian Research Promotion Agency (FFG) project 7640628 (Danio4Can). We thank Olympus for use of the Olympus Yokogawa CSU-W1 Spinning Disc Confocal microscope.

## Additional information

### Funding

| Funder | Grant reference number | Author |
|---|---|---|
| National Health and Medical Research Council | 1164734 | Benjamin M Hogan |
| National Health and Medical Research Council | 1165117 | Benjamin M Hogan |
| Austrian Research Promotion Agency | 7640628 | Martin Distel |
| National Health and Medical Research Council | 1155221 | Benjamin M Hogan |

The funders had no role in study design, data collection and interpretation, or the decision to submit the work for publication.

### Author contributions

Kazuhide S Okuda, Conceptualization, Formal analysis, Supervision, Investigation, Methodology, Writing - original draft; Mikaela S Keyser, David B Gurevich, Formal analysis, Investigation; Caterina Sturtzel, Scott Paterson, Huijun Chen, Investigation; Elizabeth A Mason, Formal analysis; Mark Scott, Nicholas D Condon, Software, Methodology; Paul Martin, Resources, Supervision; Martin Distel,

Resources, Supervision, Funding acquisition; Benjamin M Hogan, Conceptualization, Resources, Supervision, Funding acquisition, Investigation, Project administration, Writing - review and editing

Author ORCIDs
Kazuhide S Okuda (D) https://orcid.org/0000-0002-8060-0377
Martin Distel (D) http://orcid.org/0000-0001-5942-0817
Benjamin M Hogan (D) https://orcid.org/0000-0002-0651-7065

Ethics
Animal experimentation: All zebrafish work was conducted in accordance with the guidelines of the animal ethics committees at the University of Queensland (AE54297), University of Melbourne, Peter MacCallum Cancer Centre (E634 and E643), University of Bristol (3003318), and the Children's Cancer Research Institute (GZ:565304/2014/6 and GZ:534619/2014/4).

Decision letter and Author response
Decision letter https://doi.org/10.7554/eLife.62196.sa1
Author response https://doi.org/10.7554/eLife.62196.sa2

## Additional files

Supplementary files
• Transparent reporting form

Data availability
All data generated or analysed during this study are included in the manuscript and supporting files.

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
