## [Decision Letter]

**Acceptance summary:**

The study describes a new vascular endothelial specific Extracellular Signal-regulated Kinase (ERK) biosensor zebrafish line. The authors use this line to study intersegmental vessel regeneration. They observe two phases of ERK signaling, and show that first phase is dependent on calcium signaling, while the second is dependent on signalling by vascular endothelial growth factor. Overall the study is novel and highly significant, because it describes for the first time imaging of ERK dynamics during angiogenesis in live embryos. The ERK biosensor will undoubtedly be an important and useful tool for many researchers.

**Decision letter after peer review:**

Thank you for submitting your article "Live-imaging of endothelial Erk activity reveals dynamic and sequential signalling events in regenerative angiogenesis" for consideration by *eLife*. Your article has been reviewed by 3 peer reviewers, one of whom is a member of our Board of Reviewing Editors, and the evaluation has been overseen by Didier Stainier as the Senior Editor. The following individual involved in review of your submission has agreed to reveal their identity: Masahiro Shin (Reviewer #3).

The reviewers have discussed the reviews with one another and the Reviewing Editor has drafted this decision to help you prepare a revised submission.

Summary:

The manuscript by Okuda et al. describes a new zebrafish ERK biosensor line. The authors use this line to study intersegmental vessel regeneration. They observe two phases of ERK signaling, and show that first phase is dependent on Ca^2+^ signaling, while the second is dependent on VEGF signaling.

All reviewers have found that the study is novel and significant, and that materials and methodology will undoubtedly be an important and useful tool for the entire zebrafish community. Quality of experiments and images is outstanding. Although the study leaves some deeper mechanistic questions unanswered (such as, how Ca signaling leads to ERK activation), the study should be acceptable for publication in *eLife* after the following revisions.

Essential revisions:

1. It is very intriguing that ERK activation is observed in adjacent ISVs after injury. How far does ERK activation spread from the injury site? Is it observed in ISVs that are further away from the wound site?

2. The argument for the role of Vegfr signaling in promoting ongoing Erk activity is based solely on the results from SU5416 treatment. Chemical inhibitors may have non-specific effects and SU5416 can block other receptor tyrosine kinases as well. It is important to confirm this results using genetic mutant or knockdown approaches. Partial knockdown of vegfaa, and kdr or kdrl mutants which would have partial ISV sprouting defecs can be used to confirm results obtained using the chemical inhibitor.

3. The authors suggest that macrophage-secreted VEGF is responsible for continued ERK activation at 3 hpa. This should be tested directly by inhibiting macrophages using spi1b and csf3r MO approach

4. Line 396, to ask requirement of Ca^2+^ signal for the second activity of ERK at 3hpa, the authors treated amputated embryos with NF between 2.5hpa and 3hpa. It would be important to determine if Ca^2+^ between 0hpa and 15mpa is important for sustained ERK signal at 3hpa. Could the author treat amputated embryo with NF between 0 and 15mpa and measure ERK activity in ablated ECs at 3hpa?

5. According to Figure 2, tip cells show asymmetric ERK activity following cell division. Can the authors use the ERK reporter to confirm an existing proposal that ERK inhibition of Notch signal in stalk cell mediated by neighboring tip cell after their asymmetric cell division. DAPT could be useful to measure sequential ERK activity in tip and stalk cells after the cell division.

6. Please discuss how Ca^2+^ is increased in endothelial cells by citing previous studies on TRP channels or P2X receptors.

---

## [Author Response]

Essential revisions:1. It is very intriguing that ERK activation is observed in adjacent ISVs after injury. How far does ERK activation spread from the injury site? Is it observed in ISVs that are further away from the wound site?

This is a great question and we have provided further detailed analysis in response to this question. We generated data spanning a greater field of view than earlier analyses, so that we could quantify the Erk signalling response further away from the ablated vessel. We measured the response at the adjacent ISVs, the 2^nd^ and 3^rd^ ISVs away from the ablation (in an anterior direction on the embryo – see **Figure 3—figure supplement 1A**). We found that while the response of the immediately adjacent vessel is always robust, the 2^nd^ vessel displays very little Erk activation and the 3^rd^ vessel none. This new data is all now included with careful quantification in revised **Figure 3—figure supplement 1.**

Based on these new observations the response is largely in the immediately wounded vessel and immediately adjacent vessel. There is little evidence for a response further away.

2. The argument for the role of Vegfr signaling in promoting ongoing Erk activity is based solely on the results from SU5416 treatment. Chemical inhibitors may have non-specific effects and SU5416 can block other receptor tyrosine kinases as well. It is important to confirm this results using genetic mutant or knockdown approaches. Partial knockdown of vegfaa, and kdr or kdrl mutants which would have partial ISV sprouting defecs can be used to confirm results obtained using the chemical inhibitor.

We agree with the reviewer that a genetic model is required to provide further confidence in the pharmacological data. We now provide substantial new data in response to this.

We tried a few different genetic approaches here, but settled on addressing this question using a *kdrl* Crispant model. The reason for this is that we did not have a mutant line available without an extended period for importation and the Crispant was found to very reliably phenocopy previously published MO and mutant phenotype (see new **Figure 5—figure supplement 1K,L**) (Habeck et al., 2002;Covassin et al., 2006).

We generated *kdrl* Crispants and then tested the Erk signalling response to wounding at 3 hpa. We found that loss of *kdrl* clearly reduced the levels of Erk signalling (see new **Figure 5B and Figure 5figure supplement 1M-T’**), although not to the same degree as pan-VEGFR inhibitors (see **Figure 5A**). This is not unexpected as it is likely that Kdr or Flt4 may also play a role in this response (Covassin et al., 2006;Shin et al., 2016). We have also added additional explanatory text around this data and point (see **Page 11, Lines 342-350 and Pages 24,25, Lines 130-155**).

3. The authors suggest that macrophage-secreted VEGF is responsible for continued ERK activation at 3 hpa. This should be tested directly by inhibiting macrophages using spi1b and csf3r MO approach

We thank the reviewer for this suggestion and we add significant new data in response. We used the same MO approach to knock down macrophages in the embryo and determine the impact on (1) vessel regeneration at 24hpa and (2) Erk-signalling at 3hpa.

We were able to independently reproduce the work of Gurevich et al. 2018 in demonstrating that there was reduced vessel regeneration in the absence of macrophages (see new data in **Figure 3**—**figure supplement 2H-J) (Gurevich et al., 2018).** This and direct imaging of macrophages in MO knockdown embryos **(Figure 3—figure supplement 2E-G**) confirmed the model was working for additional analysis.

We then measured Erk signalling using the EKC line. We found that ablated ISV ECs maintained their Erk activity 3 hpa in the absence of macrophages (see new data in **Figure 5D and Figure 5—figure supplement 2).** This indicates that while macrophages are required for vessel regeneration, they are not the sole source of Vegfs at 3 hpa, and other local source(s) likely exist. We have also added additional explanatory text around this data and point **(see Page 11, Lines 357-360; Page 15, Lines 493-495; Page 48, 688-689**).

These new findings have prompted us to slightly revise our final model of how this mechanism works in a revised Figure 7.

4. line 396, to ask requirement of Ca^2+^ signal for the second activity of ERK at 3hpa, the authors treated amputated embryos with NF between 2.5hpa and 3hpa. It would be important to determine if Ca^2+^ between 0hpa and 15mpa is important for sustained ERK signal at 3hpa. Could the author treat amputated embryo with NF between 0 and 15mpa and measure ERK activity in ablated ECs at 3hpa?

We have now performed these experiments and found that the loss of Ca^2+^ signalling due to treatment with Nifedipine from 0-30 mpa is insufficient to impact the level of Erk-signalling at later stages (see new data in **Figure 6F and Figure 6—figure supplement 2H-N’**).

We are not sure exactly what this tells us about the functional significance of the early Ca^2+^dependent Erk burst, because its role may be more nuanced than this measurement suggests.

However, we believe it is a useful addition to the manuscript and we now include this new data in the revised **Figure 6F.** We have also added additional explanatory text around this data and point **(see Page 13, Lines 427-429; Page 24, Lines 122-126**).

5. According to Figure 2, tip cells show asymmetric ERK activity following cell division. Can the authors use the ERK reporter to confirm an existing proposal that ERK inhibition of Notch signal in stalk cell mediated by neighboring tip cell after their asymmetric cell division. DAPT could be useful to measure sequential ERK activity in tip and stalk cells after the cell division.

Unfortunately, we were unable to perform this experiment in the time period of this resubmission due purely to logistical reasons.

The requested experiment requires high-speed spinning disc microscopy. Our laboratory recently moved and we initiated the purchase of a spinning disc confocal microscope equivalent to the one used to generate most of the data in the paper. However, our purchase was frozen in early 2020 due to the impact of COVID-19. This has meant that we needed to address the reviewers concerns for this paper in the absence of the high-speed imaging capability that we had used for much of the initial submission (when we earlier had access to an Andor Dragonfly system).

The requested experiment to look into the dynamics of asymmetric Erk signalling after tip cell division is a good idea. However, we do not think that this experiment in any way changes the conclusions of the current study. It would bring in a very detailed analysis of a mechanism that we have not investigated in the original version of the manuscript. As such, and considering the reasons for our delay here, we hope the reviewers and editors will consider this request to be an extension of the current study and will understand our reasons for not providing it.

We certainly would have liked to perform this experiment and if the reviewers see it as absolutely essential for acceptance of the paper, we can still do it once our new microscope (finally) arrives in a few months (mid-2021).

6. Please discuss how Ca^2+^ is increased in endothelial cells by citing previous studies on TRP channels or P2X receptors.

We have now extended the discussion on page 15 to include a discussion of how TRP channels or P2X receptors might play a role in this process and to cite 2 relevant reviews. We think this is a good point from the reviewer and improves the discussion, but we have not gone into an in-depth discussion on this potential mechanism as we do not have data in the paper that directly relates to TRP or P2X4-7 function in zebrafish EC Ca^2+^ signalling (see **Page 16, Lines 511-515**).